# Auto-configuring Exploration-Exploitation Tradeoff in Population-Based Optimization: A Deep Reinforcement Learning Approach

## Abstract

Population-based optimization (PBO) algorithms, renowned as powerful black-box optimizers, leverage a group of individuals to cooperatively search for the optimum. The exploration-exploitation tradeoff (EET) plays a crucial role in PBO, which, however, has traditionally been governed by manually designed rules. In this paper, we propose a deep reinforcement learning-based framework that autonomously configures and adapts the EET throughout the PBO search process. The framework allows different individuals of the population to selectively attend to the global and local exemplars based on the current search state, maximizing the cooperative search outcome. Our proposed framework is characterized by its simplicity, effectiveness, and generalizability, with the potential to enhance numerous existing PBO algorithms. To validate its capabilities, we apply our framework to several representative PBO algorithms and conduct extensive experiments on the augmented CEC2021 benchmark. The results demonstrate significant improvements in the performance of the backbone algorithms, as well as favorable generalization across diverse problem classes, dimensions, and population sizes. Additionally, we provide an in-depth analysis of the EET issue by interpreting the learned behaviors of PBO.

## 1 Introduction

Using population-based Optimization (PBO) algorithms as black-box optimizers has received significant attention in the last few decades (Golovin et al., 2017; Slowik & Kwasnicka, 2020). Typically, the PBO algorithms deploy a population of individuals that work cooperatively to undertake both *exploration* (that discovers new knowledge) and *exploitation* (that advances existing knowledge), so as to make the black-box optimization problem "white" (Chen et al., 2009). Targeting global convergence to the global optimum, the *exploration-exploitation tradeoff* (EET) is the most fundamental issue in the development of PBO algorithms.

Among the extensive literature focusing on the EET issues in PBO algorithms, hyper-parameters tuning is one of the most promising way. In the vanilla PBOs (Kennedy & Eberhart, 1995; Storn & Price, 1997), the EET-related hyper-parameters such as cognitive coefficient and social coefficient in Particle Swarm Optimization (PSO) are set as static values throughout the search, necessitating laborious tuning for different problem instances. The adaptive PBOs (Liang et al., 2015; Tanabe & Fukunaga, 2014), which introduce manually designed rules to dynamically adjust EET hyper-parameters according to optimization states, soon became more flexible and powerful optimizers that dominate the performance comparisons. However, they rely heavily on human knowledge to turn raw features of the search progress into decisions on EET control, which are hence labour-intensive.

In the recent "learning to optimize" paradigm, deep reinforcement learning (DRL) based approaches have been found successful to complement or well replace conventional rule-based optimizers (Ma et al., 2022; Mischek & Musliu, 2022). When it comes to the hyper-parameters tuning for PBOs, several early attempts have already been made to control the EET hyper-parameters through DRL automatically (Tan & Li, 2021; Yin et al., 2021). Though these works have demonstrated effectiveness in automating the EET strategy design process in an end-to-end manner, they still suffer from several major limitations.

The first issue is the generalization of the learnt model. Some of the existing works stipulated DRL to be conducted online, where they trained and tested the model directly on the target problem instance, that is, their methods require (re-)training for every single problem, such as DRL-PSO (Wu & Wang, 2022), DE-DDQN (Sharma et al., 2019), DE-DQN (Tan & Li, 2021) and RLHPSDE (Tan et al., 2022). Some other works such as RLEPSO (Yin et al., 2021) trained its agent by randomly choosing a function from CEC2013 benchmark and then tested on the same benchmark functions. However,

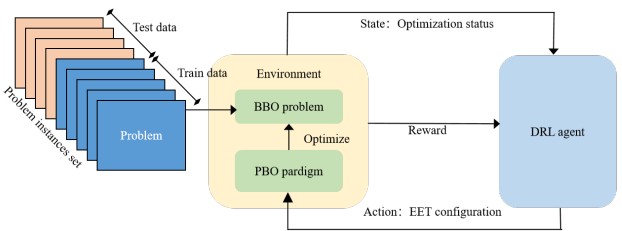

Figure 1: The overview of GLEET as an MDP.

the train-test process is also unreasonable and online in some extends. Such design, in our view, may prevent DRL from learning generalizable patterns and result in overfitting. To this end, we present the first **G**eneralizable **L**earning-based **E**xploration-**E**xploitation **T**radeoff framework, called **GLEET**, that could explicitly control the EET hyper-parameters of a given PBO algorithm to solve a class of PBOs problems via reinforcement learning. Our GLEET performs training only once on a class of black-box problems of interest, after which it uses the learned model to directly boost the backbone algorithm for other problem instances within (and even without) the same class. The overview of our GLEET is illustrated in Fig. 1. To fulfill the purpose, we formulate the GLEET as a more comprehensive Markov Decision Process (MDP) than those in the existing works, with specially designed state space, action space, and reward function to facilitate efficient learning.

The second issue arises from the oversimplified state representation and network architecture (i.e., a multi-layer perceptron), which fail to effectively extract and process the features of the EET and problem knowledge. In this paper, we design a Transformer-styled (Vaswani et al., 2017) network architecture that consists of a *feature embedding* module for feature extraction, a *fully informed encoder* for information processing amongst individuals, and an *exploration-exploitation decoder* for adjusting EET parameters. On the one hand, the transformer architecture achieves invariance in the ordering of population members, making it generalizable across problem dimension, population size and problem class. On the other hand, the proposed model allows different individuals to adaptively and dynamically attend to the knowledge of other individuals via the self-attention mechanism, so as to decide the EET behavior automatically.

Lastly, we conduct extensive experiments to verify our GLEET. Different from existing works, our augmented dataset from the CEC2021 benchmark (Mohamed et al., 2021) is larger and more comprehensive. We evaluate our GLEET by applying it to several representative PBO algorithms, i.e., the vanilla PSO (Kennedy & Eberhart, 1995) the DMSPSO (Liang & Suganthan, 2005) and the vanilla DE (Storn & Price, 1997), though we note that our GLEET has the potential to boost many other existing PBO algorithms. Results show that GLEET could significantly ameliorate the backbone algorithms, making them surpass both adaptive tuning methods and existing learning-based tuning methods. Meanwhile, our GLEET exhibits promising generalization capabilities across different problem classes, population sizes and problem dimensions. We also visualize the knowledge learnt by GLEET and interpret how it learns different EET strategies for different problem classes, which further provides insights to the PBO area.

The rest of this paper is organized as follows: Section 2 reviews how the EET issue has been addressed in traditional PBO algorithms and some recently proposed reinforcement learning-based frameworks. Section 3 introduces the preliminary concepts and notations related to DRL and PBO algorithms. In Section 4, we present the technical details of the GLEET framework, including the problem definition, network design and training process. The experiment setup and concrete experimental results are presented in Section 5.

## 2 Related Works

### 2.1 Traditional EET Methods

The vanilla PBO algorithms address the EET issue in a static manner. For example, each individual in the vanilla Particle Swarm Optimization (PSO) (Kennedy & Eberhart, 1995) controls the EET by paying equal attention to the global best experience (for exploitation) and its personal best experience (for exploration). Another example is vanilla Differential Evolution (DE) (Storn & Price, 1997), where the EET hyper-parameters $F$ (exploitation by learning from the best history) and $CR$ (exploration by perturbation or so called crossover) is pre-defined by expert knowledge to balance the EET in the optimization process. However, static EET parameters are problem agnostic hence require a tedious tuning process for each new problem, and may also limit the overall search performance.

Several adaptive PBO variants that dynamically adjust the EET-related hyper-parameters along the optimization process were then proposed to address this issue. For PSO, several early attempts focused on adaptively tuning its inertia weight (IW) hyper-parameter, such as (Shi & Eberhart, 1999; Tanweer et al., 2015; Amoshahy et al., 2016), which were then surpassed by methods of considering tuning the acceleration coefficient (AC) hyper-parameter using adaptive rules based on constriction function (Clerc & Kennedy, 2002), time-varying nonlinear function (Ratnaweera et al., 2004; Chen et al., 2018), fuzzy logic (Nobile et al., 2018), or multi-role parameter design (Xia et al., 2019). GLPSO (Gong et al., 2016) self-adapts the EET by genetic evolution. In the recent strong optimizer sDMSPSO (Liang et al., 2015), the tuning of IW and ACs are considered together into a well-known multi-swarm optimizer DMSPSO (Liang & Suganthan, 2005) to efficiently adjust EET, achieving superior performance. For DE, The parameters in mutation and crossover are the key to control its EET. Therefore, SHADE (Tanabe & Fukunaga, 2013) adopted two memories for $F$ and $Cr$ to record their statistical information instead of remembering their values. MadDE (Biswas et al., 2021) and NL-SHADE-LBC (Stanovov et al., 2022) adopted the parameter memories proposed in SHADE and employed multiple adaptive mutation and crossover operators together with the archive design. A linear population reduction method was also used to enhance their performance. Generally, most of the above methods rely heavily on human knowledge and are hence labour-intensive and vulnerable to inefficiencies.

### 2.2 Learning-based EET Methods

The EET issue was also tackled via (deep) reinforcement learning automatically. In the following, We sort out some representative works and further highlight the motivation of this study.

#### 2.2.1 For DE

In the work of Karafotias et al. (2014), Reinforcement Learning (RL) was firstly considered as a controller for PBO's hyper-parameters. The authors gave the first MDP formulation on the parameters control in PBO algorithms and utilized Q-Learning to optimize this MDP. The study was followed by successive works such as DE-DDQN (Sharma et al., 2019), DE-RLFR (Li et al., 2019) and DEDQN (Tan & Li, 2021). DE-DDQN leverages a large body of ninety-nine features to ensure successful control on selection of DE's mutation operators. The authors adopted the Deep Q-Network (DQN) (Mnih et al., 2013) to process such a high-dimensional input. DE-RLFR used simple fitness ranking information in the population as feature and a Q-table agent (Watkins & Dayan, 1992) to accomplish the similar selection mission as the DE-DDQN. Following the idea in DE-RLFR, DEDQN enriched the fitness ranking feature to a more systemic fitness landscape analysis (FLA) features and used the similar DQN agent as in DE-DDQN. Despite adopting RL for operator selection, recent works start to expand the strength of RL to hyper-parameters control in DE algorithms. One of the two representative works is LDE (Sun et al., 2021), which learns to control the step length $F$ and crossover rate $CR$ in DE algorithms. LDE adopted an LSTM network to efficiently learn from the informative features extracted from population optimization history, while a traditional version of Policy Gradient RL named REINFORCE (Williams, 1992) is employed to learn the optimal control policy. The other is RLHPSDE (Tan et al., 2022), which takes the coupling of mutation operator and its hyper-parameter value as the action space in a MDP, then leverages Q-learning to solve it.

### 2.2.2 For PSO

Samma et al. (2016) firstly explored to control the ACs in PSO by Q-learning, which inspired many followers to use deep Q-learning for topology structure selection in PSO (Xu & Pi, 2020) or parameter control in multi-objective PBOs (Liu et al., 2019). In RLEPSO (Yin et al., 2021), the authors proposed to improve EPSO (Lynn & Suganthan, 2017) by tuning its EET hyper-parameters based on the optimization schedule feature and policy gradient method. In another recent work (Wu & Wang, 2022), the DRL-PSO optimizer was presented to control the random variables in the PSO velocity update equations to address EET, where DRL-PSO outperformed the advanced adaptive method sDMSPSO on a small test dataset.

However, all the above works neglected the generalization ability of the learned model as established in the introduction. We note that in a related field of neural combinatorial optimization, researchers developed several generalizable solvers to solve a class of similar problems based on DRL (Kwon et al., 2020; Ma et al., 2022), however, their underlying MDPs and networks were specially designed for discrete optimization, making them not suitable for the hyper-parameter tuning task studied in this paper. Furthermore, the experiments of existing works are limited to small datasets (with only a dozen problem instances), which makes their performance comparison inconclusive. For example, when we train and test the DRL-PSO on much larger datasets in our experiments, we found that it could not outperform the sDMSPSO. Finally, the network architecture and input features in most existing methods are very simple, which largely limits their performance, especially when compared to the advanced adaptive PBO variants. Two recent works Meta-ES(Lange et al., 2022) and Meta-GA(Lange et al., 2023) may share the same ambition with our GLEET. They provide a brand new paradigms to meta-learn a NN parameterized population-based optimizer by Neural Evolution. They are pre-trained on a set of synthetic problems with different landscape properties and directly applied to unseen tasks. Notably, Meta-ES show remarkable robustness when zero-shot to high-dimensional continuous controlling tasks.

## 3 Preliminary and Notations

### 3.1 Deep Reinforcement Learning

DRL methods specialize in solving MDP by learning a deep network model as its decision policy (Sutton & Barto, 2018). Given an MDP formalized as $\mathcal{M} := <\mathcal{S}, \mathcal{A}, \mathcal{T}, \mathcal{R}>$, the agent obtains a state representation $s \in \mathcal{S}$ and then decides an action $a \in \mathcal{A}$ which turns state $s$ into next state $s'$ according to the dynamic of environment $\mathcal{T}(s'|s, a)$ and then gets a reward $\mathcal{R}(s, a)$. The goal of DRL is to find a policy $\pi_\theta(a|s)$ (parameterized by the deep model $\theta$) so as to optimize a discounted expected return $\mathbb{E}_{\pi_\theta}[\sum_{t=1}^{\mathcal{T}} \gamma^{t-1} \mathcal{R}(s_t, a_t)]$.

### 3.2 Attention Mechanism

In the well-known Transformer model (Vaswani et al., 2017), the attention is computed by

$$\text{Attn}(Q, K, V) = \text{softmax}\left(\frac{QK^T}{\sqrt{d_k}}\right)V \tag{1}$$

where $Q, K, V$ are the vectors of queries, keys and values respectively, and $d_k$ is the dimension of queries which plays a role as a normalizer. The Transformer consists of multiple encoders with self-Attention and decoders, both encoder and decoder use Multi-Head Attention which maps vectors into different sub-spaces for a better representation:

$$\begin{aligned} H &= \text{MHA}(Q, K, V) = \text{Concat}(H_1, H_2, \cdots, H_h)W^O \\ H_i &= \text{Attn}(QW_i^Q, KW_i^K, VW_i^V) \end{aligned} \tag{2}$$

where $h$ denotes the number of heads, $W^O \in \mathbb{R}^{hd_v \times d_s}$, $W_i^Q \in \mathbb{R}^{d_s \times d_k}$, $W_i^K \in \mathbb{R}^{d_s \times d_k}$ and $W_i^V \in \mathbb{R}^{d_s \times d_v}$. For self-attention, the $Q, K, V$ can all be derived from one input source, whereas for general attention types, $Q$ can be from a different input source other than that of $K$ and $V$.

### 3.3 Particle Swarm Optimization

This algorithm deploys a population of particles (individuals) as candidate solutions, and in each iteration $t$ it improves each particle $x_i$ as

$$x_i^{(t)} = x_i^{(t-1)} + v_i^{(t)},$$
$$v_i^{(t)} = w \times v_i^{(t-1)} + c_1 \times \text{rnd} \times \left( pBest_i^{(t-1)} - x_i^{(t-1)} \right) + c_2 \times \text{rnd} \times \left( gBest^{(t-1)} - x_i^{(t-1)} \right) \tag{3}$$

where $v_i$ is the velocity; $pBest_i$ and $gBest$ are the personal and global best positions found so far respectively; $w$ is an inertia weight; rnd returns a random number from $[0, 1]$; and $c_1$ and $c_2$ induce the EET where a large $c_1$ encourages the particle to explore different regions based on its own beliefs and a large $c_2$ forces all particles to exploit the global best one.

### 3.4 Differential Evolution

Differential Evolution (DE) proposed by Storn and Price (Storn & Price, 1997) used mutation, crossover and selection to handle a population of solution vectors iteratively and search optimal solution using the difference among population individuals. At each iteration $t$, the mutation operator is applied on individuals to generate trail vectors. One of the classic mutation operators DE/current-to-pbest/1 on the individual $x_i$ could be formulated as

$$v_i^{(t)} = x_i^{(t-1)} + F_{i,1}^{(t)} \cdot (x_{tpb}^{(t-1)} - x_i^{(t-1)}) + F_{i,2}^{(t)} \cdot (x_{r1}^{(t-1)} - x_{r2}^{(t-1)}) \tag{4}$$

where $F_{i,1}^{(t)}$ and $F_{i,2}^{(t)}$ are the step lengths controlling the variety of trials, $x_{tpb}^{(t-1)}$ is the random selected individual from the top-$p\%$ best cost individuals, $x_{r1}^{(t-1)}$ and $x_{r2}^{(t-1)}$ are two different random selected individuals. Then, the crossover is adopted to exchange values between the parent $x_i^{(t-1)}$ and the trial individual $v_i^{(t)}$ to produce an offspring $u_i^{(t)}$:

$$u_{i,j}^{(t)} = \begin{cases} v_{i,j}^{(t)} & \text{if rand}[0,1] \leq Cr_i^{(t)} \text{ or } j = jrand \\ x_{i,j}^{(t-1)} & \text{otherwise} \end{cases} \tag{5}$$

where $u_{i,j}^{(t)}$ is the $j$-th value of the individual $u_i^{(t)}$ and so do the items for $x_i^{(t-1)}$ and $v_i^{(t)}$. The $Cr_i^{(t)}$ is the crossover rate and $jrand$ is an index to ensure the difference between $u_i^{(t)}$ and $x_i^{(t)}$. Finally, the selection method eliminates those individuals with worse fitness than their parents.

## 4 Methodology of GLEET

### 4.1 MDP Formulation

Given a population $P$ with $N$ individuals, an PBO algorithm $\Lambda$, and a problem set $D$, we formulate the dynamic hyper-parameter tuning as an MDP:

$$\mathcal{M} :=< \mathcal{S} = \{s_i\}_{i=1}^N, \mathcal{A} = \{a_i\}_{i=1}^N, \mathcal{T}, \mathcal{R} > \tag{6}$$

where state $\mathcal{S}$ and action $\mathcal{A}$ take all individuals in the Population $P$ into account. Each $a_i \in \mathbb{R}^M$ denotes the choice of hyper-parameters for the $i$-th individual in $\Lambda$, where $M$ denotes the number of hyper-parameters to be controlled. For example, in DE/current-to-pbest/1 (Zhang & Sanderson, 2009), the hyper-parameters $F_1$, $F_2$ and $Cr$ need to be determined. The transition function $\mathcal{T} : \mathcal{A} \times \Lambda \times P \to P$ denotes evolution of population $P$ through algorithm $\Lambda$ with hyper-parameters $\mathcal{A}$. The reward function $\mathcal{R} : \mathcal{S} \times \mathcal{A} \times D \to \mathbb{R}^+$ measures the improvement in one optimization step brought by dynamic hyper-parameter settings.

While optimizing the hyper-parameters of $\Lambda$ on a single problem instance may yield satisfactory results, it can limit the generalization performance on unseen problems. To address this, we construct a problem set

$D$ comprising $K$ problems (detailed in Section 5.1.1) to facilitate generalization. Correspondingly, the DRL agent targets at the optimal policy $\pi_{\theta^*}$ that controls the dynamic hyper-parameters for $\Lambda$ to maximize the expected accumulated reward over all the problems $D_k \in D$ as

$$\theta^* := \underset{\theta \in \Theta}{\arg\max} \frac{1}{K} \sum_{k=1}^{K} \sum_{t=1}^{T} \gamma^{t-1} \mathcal{R}(S^{(t)}, A^{(t)} | D_k). \tag{7}$$

This approach ensures robust performance across a diverse range of problem instances and promotes generalization capabilities.

### 4.1.1 State

We provide four principles for the GLEET state space design: a) it should describe the state of every individual in the population; b) it should be able to characterize the optimization progress and the EET behavior of the current population; c) it should be compatible across different kinds of PBO algorithms and achieve generalization requirement; and d) it should be convenient to obtain in each optimization step of the PBO process.

Following these four principles, we use a $K$-dimensional vector to define the state $s_i$ for each individual in PBO algorithm at time step $t$. The information required to calculate this state vector includes the global best individual $gBest$ in the population, each individual's historical best information $pBest_i$ and current position $x_i$, as well as their evaluation values $f(gBest)$, $f(pBest_i)$ and $f(x_i)$. For most of PBO algorithms, information above is compatible and easy to obtain, which makes GLEET a generic paradigm for boosting many kinds of PBO algorithms. Computational detail is shown in Eq. (8), $K = 9$ where we compute: $s_{i,\{1,2\}}$ as the search progress w.r.t. the $gBest$ value and the currently consumed number of fitness evaluations ($FE$); $s_{i,\{3,4\}}$ as the stagnation status w.r.t. the number of rounds $z(\cdot)$ for which the algorithm failed to find a better $gBest$ or $pBest_i$, normalized by the total rounds $T_{\max}$; $s_{i,\{5,6\}}$ as the difference in evaluation values between individuals and $gBest$ or $pBest_i$, normalized by the initial best value; $s_{i,\{7,8\}}$ as the Euclidean distance between particles and $gBest$ or $pBest_i$, normalized by the diameter of the search space; $s_{i,\{9\}}$ as the cosine function value of the angle formed by the current particle to the $gBest$ and $pBest_i$.

$$s_{i,\{1,2\}} = \left\{ \frac{f(gBest)}{f(gBest^{(0)})}, \frac{FE_{\max} - FE}{FE_{\max}} \right\}, s_{i,\{3,4\}} = \left\{ \frac{z(gBest)}{T_{\max}}, \frac{z(pBest_i)}{T_{\max}} \right\},$$

$$s_{i,\{5,6\}} = \left\{ \frac{f(x_i) - f(gBest)}{f(gBest^{(0)})}, \frac{f(x_i) - f(pBest_i)}{f(pBest_i^{(0)})} \right\}, s_{i,\{7,8\}} = \left\{ \frac{||x_i - gBest||}{diameter}, \frac{||x_i - pBest_i||}{diameter} \right\}, \tag{8}$$

$$s_{i,\{9\}} = \left\{ \cos\left(\angle\left(gBest - x_i, pBest_i - x_i\right)\right) \right\},$$

Note that we also calculate the above $K$ features for the global best position and $N$ personal best positions to learn embeddings for them (will be used in the decoder). We name the $N \times K$ state features of $\{x_i\}_{i=1}^{N}$ as the *population features*, the $1 \times K$ state features of $gBest$ as the *exploitation features*, and the $N \times K$ state features of $\{pBest_i\}_{i=1}^{N}$ as the *exploration features*, respectively. These three parts of features together composite the state of GLEET's MDP. We again emphasize that this state design is generic across different PBO algorithms which makes GLEET generalizable for a large body of PBO algorithms.

### 4.1.2 Action

Since the hyper-parameters in most PBO algorithms are continuous, and discretizing the action space may damage the action structure or cause the curse of dimensionality issue (Lillicrap et al., 2015), GLEET prefers continuous action space that jointly controls all $N$ individuals' choices of hyper-parameters $(a_1^{(t)}, a_2^{(t)}, \cdots, a_N^{(t)})$, where $a_i^{(t)}$ denotes $M$ hyper-parameters for individual $i$ at time step $t$. Concretely, the action probability $\Pr(a)$ is a multiplication of normal distributions as follows,

$$\Pr(a) = \prod_{i=1}^{N} \prod_{m=1}^{M} p(a_i^m), \quad a_i^m \sim \mathcal{N}(\mu_i^m, \sigma_i^m) \tag{9}$$

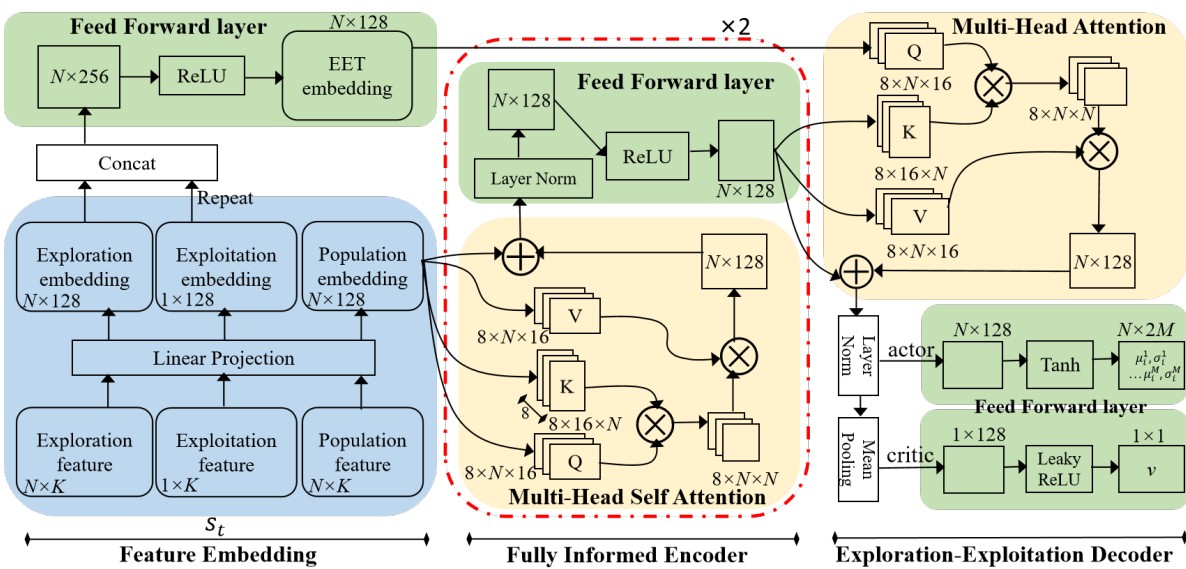

Figure 2: Illustration of our network design. The network begins by embedding the state feature into two components: the EET embedding and the population embedding. Next, a Fully Informed Encoder is employed to attend the population embedding to the individual level. Finally, the individual's EET configuration is determined by decoding the information from the EET embedding using the Exploration-Exploitation Decoder.

where $\mu_i^m$ and $\sigma_i^m$ are controlled by DRL agent. Generally, our policy network outputs $N \times M$ pairs of $(\mu_i^m, \sigma_i^m)$, where each $(\mu_i^m, \sigma_i^m)$ is used to sample a parameter to control the EET of each individual respectively.

### 4.1.3 Reward

We consider the reward function as follows,

$$r^{(t)} = \frac{f(gBest^{(t-1)}) - f(gBest^{(t)})}{f(gBest^{(0)})} \tag{10}$$

where the reward is positive if and only if a better solution is found. It is worth mentioning that there are several practical reward functions proposed previously. Yin et al. (Yin et al., 2021) rewards an improvement of solution 1 otherwise $-1$. Sun et al. (Sun et al., 2021) calculates a relative improvement between steps as reward function. Wu et al. (Wu & Wang, 2022) has a similar form with our reward function but permits negative reward. We conduct comparison experiments on these reward functions and ours, It turns out in our experiment setting, reward function proposed in Eq. (10) stands out. Results can be found in Section 5.5.

### 4.2 Network Design

As depicted in Fig. 2, fed with the state features, our actor $\pi_\theta$ first generates a set of population embeddings (based on population features) and a set of EET embeddings (based on exploitation and exploration features). The former is further improved by the fully informed encoders. These embeddings are then fed into the designed decoder to specify an action. We also consider a critic $v_\phi$ to assist the training of the actor.

### 4.2.1 Feature embedding

We linearly project raw state features from Section 4.1 into three groups of 128-dimensional embeddings, i.e., exploration embeddings (EREs) $\{h_i\}_{i=1}^N$, exploitation embedding (EIE) $\{g\}$, and population embeddings (PEs) $\{e_i\}_{i=1}^N$. Then we concatenate each ERE with the EIE to form 256-dimensional vectors $\{h_i \| g\}_{i=1}^N$ which are then processed by an MLP with structure ($256 \times 256 \times 128$, ReLU is used by default) to obtain the EET embeddings (EETs) $\{EE_i\}_{i=1}^N$ that summarize the current EET status.

### 4.2.2 Fully informed encoder

The encoders mainly follow the design of the original Transformer, except that the positional encoding is removed and layer normalization (Ba et al., 2016) is used instead of batch normalization (Ioffe & Szegedy, 2015). There are three main factors that lead us to favor full attention over sparse attention (Zaheer et al., 2020): a) the number of embeddings (i.e., population size) in our task is relatively small compared with the number of word embeddings in language processing, negating the need for sparse attention; and b) it gives the network maximum flexibility without any predefined restrictions on sparsity and hence via the attention scores, the network could automatically adjust the topology between individuals.

Specifically, our encoders update the population embeddings by

$$\hat{e} = \text{LN}(e^{(l-1)} + \text{MHA}^{(l)}(Q^{(l)}, K^{(l)}, V^{(l)})), \quad e^{(l)} = \text{LN}(\hat{e} + \text{FF}^{(l)}(\hat{e})) \tag{11}$$

where $Q^{(l)}$, $K^{(l)}$ and $V^{(l)}$ are transformed from population embeddings $e^{(l-1)}$ and $l = \{1, 2\}$ is the layer index (we stack 2 encoders). The initial condition $e^{(0)}$ is the population embeddings embedded from the population features as shown in Figure 2. The Feed Forward (FF) layer is an MLP with structure ($128 \times 256 \times 128$). The final output by our encoders is the fully informed population embeddings (FIPEs) $\{e_i^{(2)}\}_{i=1}^N$.

### 4.2.3 Exploration-exploitation decoder

Given the EETs and FIPEs, the exploration-exploitation decoder outputs the joint distribution of hyper-parameter settings for each particle. The EET control logits $H$ ($\{H_i\}_{i=0}^N$) are calculated as:

$$\hat{H} = \text{LN}(e^{(2)} + \text{MHA}^d(Q^d, K^d, V^d)), \quad H = \text{ReLU}(\text{FF}^{\text{logit}}(\text{LN}(\hat{H} + \text{FF}^d(\hat{H})))) \tag{12}$$

where we let $Q^d$ from EETs, and $K^d, V^d$ from FIPEs (different from the self-attention in the encoders); $\text{FF}^d$ is with structure ($128 \times 256 \times 128$); and $\text{FF}^{\text{logit}}$ is with structure ($128 \times 128$). We then linearly transform each $\{H_i\}_{i=1}^N$ from 128 to $2M$ scalars which are then passed through the Tanh function and scaled to range $[\mu_{\min}^m, \mu_{\max}^m]$ and $[\sigma_{\min}^m, \sigma_{\max}^m]$, respectively, in order to obtain the $a_i^m = \mathcal{N}(\mu_i^m, \sigma_i^m)$ for each individual. We set all $[\mu_{\min}, \mu_{\max}] = [0, 1]$ and all $[\sigma_{\min}, \sigma_{\max}] = [0.01, 0.7]$. To be specific, for PSO, we let $M = 1$ to control $c_{1,i}$ in Eq. (3) for each particle $i$ (leaving $c_{2,i} = 4 - c_{1,i}$ by the suggestion of (Wang et al., 2018). For DE, we let $M = 3$ to control $F_{1,i}$, $F_{2,i}$ and $Cr_i$ in Eq. (4) and Eq. (5).

### 4.2.4 Critic network

The critic $v_\phi$ shares the EET control logits $H$ from the actor and has its own output layers. Specifically, it performs a mean pooling for the logits $\bar{H} = \sum_{i=1}^N H_i$, and then processes it using an MLP with structure ($128 \times 64 \times 32 \times 1$ and LeakyReLU activation) to obtain the value estimation.

### 4.3 Training

Our GLEET agent can be trained via any off-the-shelf reinforcement learning algorithm, and we use the $T$-step PPO (Schulman et al., 2017) in this paper. A training dataset containing a class of similar black-box problem instances is generated before training (details in Section 5.1.1), from which a small batch of problem instances is randomly sampled on the fly during training (which is different from previous works that only leverages one single instance). Given the batch, we initialize a population of individuals according to the backbone PBO algorithm for each problem instance and then let the on-policy PPO algorithm gather trajectories while updating the parameters of the actor and the critic networks defined in Section 4.2. We alternate between sampling trajectory $\mathcal{T}$ by $T$ time steps and update $\kappa$ times of the network parameters. The learned model will be directly used to infer the EET control for other unseen problem instances.

## 5 Experiments

Our experiments research the following questions:

- RQ1: How good is the control performance of GLEET against the previous static, adaptive tuning and DRL-tuning methods, and whether GLEET can be broadly used for enhancing different PBO algorithms?
- RQ2: Does GLEET possess the generalization ability on the unseen problems instances, dimensions, and population sizes?
- RQ3: Can the learned behavior of GLEET be interpreted and recognized by human experts?
- RQ4: How crucial are reward design, the EET embeddings and the attention modules in our GLEET implementation?

To investigate RQ1 to RQ4, we firstly instantiate GLEET to PSO and DE and compare their optimization performance with several competitors such as PSO/DE's original version, adaptive variants and reinforcement learning-based versions, see Section 5.2. We then test the zero-shot generalization ability of GLEET by directly applying the trained agent to unseen settings, see Section 5.3. Next, we visualize the controlled EET patterns of GLEET and the decision layer of GLEET's network to interpret the learned knowledge, see Section 5.4. At last, for answering RQ4, we conduct ablation study on the reward mechanism, the EET embeddings and the attention modules, see Section 5.5.

## 5.1 Experimental Setup

### 5.1.1 Dataset augmentation

The CEC 2021 numerical optimization test suite by Ali et al. (Mohamed et al., 2021) consists of ten challenging black-box optimization problems. The $f_1$, $f_2$, $f_3$ and $f_4$ are single problems which are not any problems' hybridization or composition. For example, Bent Cigar function ($f_1$) has the form as $f(\mathbf{x}) = x_1^2 + 10^6 \sum_{i=2}^{d} x_i^2$, where $d$ is the dimension of $\mathbf{x}$. The $f_5$, $f_6$ and $f_7$ hybridize some basic functions, resulting more difficult problem because of solution space coupling. The $f_8$, $f_9$ and $f_{10}$ are linear compositions of some single problems, resulting more difficult problem because of fitness space coupling. Each problem has the form $F(x) = f(M^T(x - o))$, where $f$ is the objective function, $o$ is shift vector of the global optimum position, $M$ is the rotation matrix of the entire problem space, and we have the optimal cost as 0 for all the cases. We then augment these ten problems to their problem sets by adding different shift and rotation. To be concrete, for a function $f_i$, we firstly generate a random shift with a range $[o_{min}, o_{max}]$ which assures that the optimum of the shifted problem instances will not escape from the present search space. Then we generate a random standard orthogonal matrix $M$ (Schmidt, 1907) and then apply it to the shifted function instance. By repeatedly applying these two transformations above, we can get an augmented problem sets $D$. As one purpose of this study is to perform generalizable learning on a class of problems, we construct an augmented dataset $D$ of a large number of benchmark problem instances. Specifically, for each problem class in the CEC 2021, a total number of 1152 instances are randomly generated and divided into training and testing sets with a partition of 128:1024 (note that we use small training but large testing sets to fully validate the generalization). We train one GLEET agent per problem class and also investigate the performance of GLEET if trained on a mixed dataset containing all the problem classes (all ten functions with different $M$ and $o$ in a training set), denoted as $f_{mix}$.

### 5.1.2 Competitors

As established, existing studies control the EET in static, adaptive, and learning-based manners. We choose competitors for GLEET on PSO and GLEET on DE from these three categories. For PSO, we choose the vanilla PSO (Shi & Eberhart, 1998) as static baseline, the PSO variants DMSPSO (Liang & Suganthan, 2005), sDMSPSO (Liang et al., 2015), GLPSO (Gong et al., 2016) as adaptive baselines, and DRL-PSO (Wu & Wang, 2022) and RLEPSO (Yin et al., 2021) as the learning-based competitors. We instantiate GLEET to the vanilla PSO and DMSPSO for comparison, denoted as GLEET-PSO and GLEET-DMSPSO. For DE, we choose the DE/current-to-pbest/1 (Zhang & Sanderson, 2009) as static baseline, the DE variants MadDE (Biswas et al., 2021), NL-SHADE-LBC (Stanovov et al., 2022) as adaptive baselines, and DE-DDQN (Sharma et al., 2019), DEDQN (Tan & Li, 2021), LDE (Sun et al., 2021), RLHPSDE (Tan et al., 2022) as the learning-based competitors. We instantiate GLEET on DE/current-to-pbest/1 to join the comparison, denoted as GLEET-DE. Baselines GLPSO, MadDE, NL-SHADE-LBC, LDE, DE-DDQN are

Table 1: Numerical comparison results for PSO algorithms on $10D$ problems, where the mean, standard deviations and performance ranks are reported (with the best mean value on each problem highlighted in **bold**).

| Type | Static | | Adaptive | | | | | | DRL | | | | | | | |
|---|---|---|---|---|---|---|---|---|---|---|---|---|---|---|---|---|
| Algorithm | PSO | | DMSPSO | | sDMSPSO | | GLPSO | | DRL-PSO | | RLEPSO | | GLEET-PSO | | GLEET-DMSPSO | |
| Metrics | Mean (Std) | Rank | Mean (Std) | Rank | Mean (Std) | Rank | Mean (Std) | Rank | Mean (Std) | Rank | Mean (Std) | Rank | Mean (Std) | Rank | Mean (Std) | Rank |
| $f_1$ | 7.071E+06 (1.104E+07) | 7 | 5.903E+04 (8.036E+04) | 4 | 6.408E+03 (1.670E+04) | 2 | 1.646E+04 (1.652E+04) | 3 | 1.296E+07 (1.513E+07) | 8 | 5.418E+06 (7.420E+06) | 6 | 2.748E+06 (4.205E+06) | 5 | **2.471E+02** (7.676E+02) | 1 |
| $f_2$ | 8.428E+02 (2.716E+02) | 8 | 3.376E+02 (1.549E+02) | 2 | 5.232E+02 (1.722E+02) | 5 | 3.750E+02 (2.264E+02) | 3 | 6.253E+02 (2.166E+02) | 6 | 7.188E+02 (2.405E+02) | 7 | 5.105E+02 (1.776E+02) | 4 | **2.440E+02** (1.396E+02) | 1 |
| $f_3$ | 2.934E+01 (7.850E+00) | 7 | 1.563E+01 (2.546E+00) | 2 | 2.678E+01 (6.625E+00) | 6 | 1.781E+01 (3.677E+00) | 3 | 2.174E+01 (5.245E+00) | 5 | 3.022E+01 (7.809E+00) | 8 | 2.120E+01 (4.705E+00) | 4 | **1.498E+01** (2.357E+00) | 1 |
| $f_4$ | 2.099E+00 (1.235E+00) | 6 | 1.375E+00 (7.083E-01) | 4 | 5.948E-01 (2.471E-01) | 2 | 8.750E-01 (4.023E-01) | 3 | 7.949E+00 (9.389E+00) | 8 | 3.519E+00 (1.980E+00) | 7 | 1.422E+00 (7.776E-01) | 5 | **5.816E-01** (2.210E-01) | 1 |
| $f_5$ | 3.395E+03 (5.793E+03) | 6 | 4.282E+02 (2.221E+02) | 3 | **3.714E+02** (2.183E+02) | 1 | 2.223E+02 (1.614E+03) | 6 | 1.048E+04 (2.057E+04) | 8 | 2.119E+03 (2.912E+03) | 5 | 1.847E+03 (2.552E+03) | 4 | 3.716E+02 (1.866E+02) | 2 |
| $f_6$ | 8.443E+01 (5.844E+01) | 8 | 2.870E+01 (2.146E+01) | 4 | 1.981E+01 (1.704E+01) | 2 | 2.768E+01 (2.560E+01) | 3 | 6.379E+01 (4.684E+01) | 6 | 8.270E+01 (5.471E+01) | 7 | 4.449E+01 (3.381E+01) | 5 | **1.300E+01** (1.020E+01) | 1 |
| $f_7$ | 1.722E+03 (3.667E+03) | 5 | 2.134E+02 (1.462E+02) | 3 | 1.709E+02 (1.722E+02) | 2 | 8.152E+02 (5.051E+02) | 5 | 1.431E+03 (1.850E+03) | 7 | 1.151E+03 (9.557E+02) | 6 | 4.977E+02 (3.549E+02) | 4 | **1.302E+02** (8.489E+01) | 1 |
| $f_8$ | 3.376E+02 (2.902E+02) | 8 | 9.572E+01 (3.891E+01) | 2 | 1.495E+02 (8.792E+01) | 5 | 1.441E+02 (9.497E+01) | 4 | 2.120E+02 (1.744E+02) | 7 | 1.686E+02 (1.219E+02) | 6 | 1.096E+02 (3.924E+01) | 3 | **7.216E+01** (3.555E+01) | 1 |
| $f_9$ | 2.370E+02 (5.916E+01) | 8 | 1.690E+02 (4.759E+01) | 4 | 1.392E+02 (5.788E+01) | 2 | 1.953E+02 (2.989E+01) | 6 | 2.010E+02 (6.135E+01) | 7 | 1.862E+02 (6.679E+01) | 5 | 1.665E+02 (6.137E+01) | 3 | **1.202E+02** (2.016E+01) | 1 |
| $f_{10}$ | 2.227E+02 (3.946E+01) | 8 | 2.035E+02 (2.030E+01) | 4 | 1.955E+02 (2.287E+01) | 3 | 2.166E+02 (2.221E+01) | 5 | 2.201E+02 (3.665E+01) | 7 | 2.199E+02 (3.300E+01) | 6 | 1.882E+02 (4.127E+01) | 2 | **1.682E+02** (1.648E+01) | 1 |
| $f_{mix}$ | 8.445E+05 (1.090E+06) | 7 | 1.612E+02 (7.617E+01) | 2 | 5.105E+02 (9.813E+02) | 4 | 4.747E+02 (3.273E+02) | 3 | 3.184E+05 (5.086E+05) | 6 | 1.156E+06 (1.380E+06) | 8 | 4.225E+04 (2.068E+04) | 5 | **1.364E+02** (5.839E+01) | 1 |
| Avg Rank | 7.45 | | 3.09 | | 3.09 | | 4.00 | | 6.82 | | 6.45 | | 4.00 (↑ **48%**) | | 1.09 (↑ **35%**) | |

tested based on the original source code provided by the original authors. For the other baselines (including DMSPSO, sDMSPSO, DRL-PSO, RLEPSO, DE-DQN and RLHPSDE) that do not have open-source code available, we implement them strictly following the pseudocodes in their original manuscripts. For all baselines, we have followed their recommended hyper-parameter settings and ensured that the code and settings we used could achieve similar performance on the benchmark they used in their original paper. The learning-based competitors are trained on the same training sets as GLEET methods.

Table 2: Numerical comparison results for PSO algorithms on $30D$ problems, where the mean, standard deviations and performance ranks are reported (with the best mean value on each problem highlighted in **bold**).

| Type | Static | | Adaptive | | | | | | DRL | | | | | | | |
|---|---|---|---|---|---|---|---|---|---|---|---|---|---|---|---|---|
| Algorithm | PSO | | DMSPSO | | sDMSPSO | | GLPSO | | DRL-PSO | | RLEPSO | | GLEET-PSO | | GLEET-DMSPSO | |
| Metrics | Mean (Std) | Rank | Mean (Std) | Rank | Mean (Std) | Rank | Mean (Std) | Rank | Mean (Std) | Rank | Mean (Std) | Rank | Mean (Std) | Rank | Mean (Std) | Rank |
| $f_1$ | 2.261E+08 (2.343E+08) | 6 | 4.491E+07 (2.733E+07) | 4 | 1.319E+06 (1.444E+06) | 2 | **4.568E+05** (4.164E+05) | 1 | 9.316E+08 (9.470E+08) | 8 | 2.616E+08 (1.591E+08) | 7 | 1.336E+08 (1.512E+08) | 5 | 9.731E+06 (6.314E+06) | 3 |
| $f_2$ | 4.022E+03 (6.617E+02) | 8 | 3.114E+03 (5.146E+02) | 3 | 3.248E+03 (4.834E+02) | 4 | 2.841E+03 (7.063E+02) | 2 | 3.759E+03 (6.517E+02) | 6 | 3.998E+03 (3.071E+02) | 7 | 3.442E+03 (5.692E+02) | 5 | **2.648E+03** (5.041E+02) | 1 |
| $f_3$ | 2.070E+02 (5.150E+01) | 7 | 1.006E+02 (1.475E+01) | 3 | 2.772E+02 (5.886E+01) | 8 | 6.921E+01 (1.156E+01) | 2 | 1.486E+02 (4.956E+01) | 4 | 1.884E+02 (2.173E+01) | 6 | 1.552E+02 (3.701E+01) | 5 | **6.613E+01** (1.146E+01) | 1 |
| $f_4$ | 2.762E+01 (1.660E+01) | 6 | 1.066E+01 (3.076E+00) | 3 | 5.738E+01 (4.634E+01) | 7 | **6.644E+00** (2.526E+00) | 1 | 1.237E+03 (2.401E+03) | 8 | 1.860E+01 (1.087E+01) | 5 | 1.680E+01 (9.320E+00) | 4 | 9.433E+00 (2.633E+00) | 2 |
| $f_5$ | 3.972E+05 (4.395E+05) | 7 | 7.148E+04 (8.678E+04) | 4 | **2.653E+03** (1.441E+032) | 1 | 3.050E+04 (1.763E+04) | 2 | 7.231E+05 (1.119E+06) | 8 | 3.893E+05 (1.578E+05) | 6 | 1.868E+05 (1.921E+05) | 5 | 6.751E+04 (5.630E+04) | 3 |
| $f_6$ | 9.998E+02 (3.003E+02) | 8 | 3.778E+02 (1.581E+02) | 3 | 5.722E+02 (1.994E+02) | 4 | 3.162E+02 (1.816E+02) | 2 | 9.500E+02 (3.291E+02) | 7 | 8.127E+02 (1.257E+02) | 6 | 6.880E+02 (2.149E+02) | 5 | **3.114E+02** (1.159E+02) | 1 |
| $f_7$ | 1.222E+05 (1.469E+05) | 7 | 2.110E+04 (2.143E+04) | 4 | **1.387E+04** (6.703E+02) | 1 | 2.011E+04 (1.024E+04) | 3 | 2.680E+05 (4.235E+05) | 8 | 4.652E+04 (4.199E+04) | 5 | 4.930E+04 (4.420E+04) | 6 | 1.817E+04 (1.299E+04) | 2 |
| $f_8$ | 3.659E+03 (1.379E+03) | 7 | 1.009E+03 (6.969E+02) | 2 | 2.185E+03 (1.061E+031) | 4 | 1.774E+03 (1.002E+03) | 3 | 3.196E+03 (1.347E+03) | 6 | 3.867E+03 (6.564E+02) | 8 | 2.414E+03 (1.283E+03) | 5 | **8.505E+02** (5.703E+02) | 1 |
| $f_9$ | 7.835E+02 (1.345E+02) | 8 | 4.079E+02 (2.885E+02) | 3 | 5.335E+02 (5.327E+01) | 4 | 4.069E+02 (1.860E+02) | 2 | 7.194E+02 (1.191E+02) | 6 | 7.323E+02 (5.245E+01) | 7 | 6.377E+02 (9.189E+01) | 5 | **3.941E+02** (2.848E+01) | 1 |
| $f_{10}$ | 4.158E+02 (7.642E+01) | 6 | 2.984E+02 (3.578E+01) | 4 | 2.765E+02 (2.845E+01) | 2 | 2.836E+02 (3.336E+01) | 3 | 6.963E+02 (3.283E+02) | 8 | 4.222E+02 (3.785E+01) | 7 | 3.930E+02 (7.612E+01) | 5 | **2.608E+02** (2.852E+01) | 1 |
| $f_{mix}$ | 2.218E+07 (3.048E+07) | 7 | 5.774E+06 (3.103E+06) | 4 | 6.366E+04 (6.227E+04) | 3 | 5.313E+04 (4.039E+04) | 2 | 1.948E+07 (3.566E+07) | 6 | 4.438E+07 (3.849E+07) | 8 | 7.452E+06 (6.845E+06) | 5 | **4.894E+04** (4.098E+04) | 1 |
| Avg Rank | 7.00 | | 3.36 | | 3.64 | | 2.09 | | 6.82 | | 6.54 | | 5.00 (↑ **35%**) | | 1.55 (↑ **28%**) | |

### 5.1.3 Settings

Following the suggestion of Ali et al. (Mohamed et al., 2021), the maximum number of function evaluations ($maxFEs$) for all algorithms in experiments is set to $2 \times 10^5$ for $10D$ and $10^6$ for $30D$ problems. The search space of all problems is a real-parameter space $[o_{min}, o_{max}]^d$ with $o_{min} = -100$ and $o_{max} = 100$. We accelerate the learning and testing process through batching problem instances in the training set and testing set, with $batch\_size = 16$. For the optimization of each instance, GLEET agent acquires states from the population with $N = 100$ and samples hyper-parameters. For every $T = 10$ steps of optimization, the agent updates its network for $\kappa = 3$ steps in an PPO algorithmic manner. The training runs $MaxEpoch = 100$ with the learning rates $lr = 4e - 5$ and decays to $1e - 5$ at the end for both policy net and critic net. For the fairness of comparison, all learning based algorithms update their model by equal steps. The rest

Table 3: Numerical comparison results for DE algorithms on 10$D$ problems, where the mean, standard deviations and performance ranks are reported (with the best mean value on each problem highlighted in **bold**).

| Type | Static | | Adaptive | | | | DRL | | | | | | | | |
|---|---|---|---|---|---|---|---|---|---|---|---|---|---|---|---|
| Algorithm | DE | | MadDE | | NL-SHADE-LBC | | DE-DDQN | | DE-DQN | | LDE | | RLHPSDE | | GLEET-DE | |
| Metrics | Mean (Std) | Rank | Mean (Std) | Rank | Mean (Std) | Rank | Mean (Std) | Rank | Mean (Std) | Rank | Mean (Std) | Rank | Mean (Std) | Rank | Mean (Std) | Rank |
| $f_1$ | 6.423E+06 (6.523E+06) | 8 | 7.951E-09 (1.447E-09) | 3 | 6.545E-09 (1.017E-09) | 2 | 5.631E+06 (4.565E+06) | 6 | 5.723E+06 (5.065E+06) | 7 | 3.855E+04 (6.856E+04) | 4 | 6.453E+04 (1.142E+05) | 5 | **1.136E-09** (**1.789E-09**) | 1 |
| $f_2$ | 7.699E+02 (2.037E+02) | 8 | 3.911E+02 (1.108E+02) | 2 | **3.843E+02** (**1.198E+02**) | 1 | 7.534E+02 (1.867E+02) | 7 | 6.811E+02 (2.134E+02) | 6 | 5.999E+02 (1.190E+02) | 4 | 6.358E+02 (1.842E+02) | 5 | 4.522E+02 (1.403E+02) | 3 |
| $f_3$ | 2.099E+01 (2.393E+00) | 5 | 1.793E+01 (2.489E+00) | 2 | 1.821E+01 (2.430E+00) | 3 | 3.052E+01 (2.355E+00) | 7 | 3.242E+01 (3.564E+00) | 8 | 1.903E+01 (1.675E+00) | 4 | 2.101E+01 (1.716E+00) | 6 | **1.746E+01** (**2.419E+00**) | 1 |
| $f_4$ | 1.329E+00 (3.606E-01) | 8 | 5.161E-01 (1.722E-01) | 2 | 5.234E-01 (1.587E-01) | 3 | 1.265E+00 (3.545E-01) | 7 | 1.054E+00 (3.455E-01) | 6 | 6.172E-01 (2.465E-01) | 4 | 7.886E-01 (2.646E-01) | 5 | **4.990E-01** (**1.629E-01**) | 1 |
| $f_5$ | 1.747E+02 (9.863E+01) | 7 | 1.789E+01 (2.175E+01) | 2 | **1.711E+01** (**2.543E+01**) | 1 | 1.652E+02 (9.665E+01) | 6 | 1.795E+02 (1.005E+02) | 8 | 3.719E+01 (3.787E+01) | 4 | 3.831E+01 (3.857E+01) | 5 | 3.573E+01 (3.274E+01) | 3 |
| $f_6$ | 1.303E+01 (6.778E+00) | 6 | 5.160E+00 (3.277E+00) | 3 | 3.735E+00 (2.178E+00) | 2 | 1.132E+02 (9.545E+00) | 7 | 1.359E+02 (9.426E+00) | 8 | 6.269E+00 (5.417E+00) | 4 | 6.569E+00 (5.851E+00) | 5 | **6.671E-01** (**1.448E+00**) | 1 |
| $f_7$ | 4.726E+01 (4.073E+01) | 8 | 9.162E+00 (9.163E+00) | 3 | 8.424E+00 (8.127E+00) | 2 | 3.656E+01 (3.212E+01) | 6 | 4.598E+01 (3.532E+01) | 7 | 1.278E+01 (1.552E+01) | 4 | 1.953E+01 (2.354E+01) | 5 | **8.226E+00** (**1.497E+01**) | 1 |
| $f_8$ | 6.803E+01 (1.136E+01) | 4 | 5.757E+01 (1.659E+01) | 3 | 5.014E+01 (1.451E+01) | 2 | 6.956E+01 (1.215E+01) | 5 | 7.083E+01 (1.125E+01) | 6 | 8.867E+01 (1.832E+01) | 8 | 8.241E+01 (1.687E+01) | 7 | **5.349E+01** (**1.627E+01**) | 1 |
| $f_9$ | 1.765E+02 (2.277E+01) | 8 | **8.424E+01** (**3.828E+01**) | 1 | 8.765E+01 (3.934E+01) | 2 | 1.533E+02 (2.545E+01) | 5 | 1.456E+02 (1.916E+01) | 4 | 1.612E+02 (3.313E+01) | 6 | 1.655E+02 (3.438E+01) | 7 | 1.249E+02 (5.208E+01) | 3 |
| $f_{10}$ | 2.370E+02 (1.769E+01) | 8 | **1.865E+02** (**1.008E+01**) | 1 | 1.894E+02 (1.132E+01) | 2 | 2.342E+02 (1.615E+01) | 7 | 2.245E+02 (1.531E+01) | 6 | 1.994E+02 (1.284E+01) | 4 | 2.124E+02 (1.142E+01) | 5 | 1.937E+02 (1.211E+01) | 3 |
| $f_{mix}$ | 1.733E+02 (7.912E+01) | 7 | 7.570E+01 (2.311E+01) | 2 | **7.413E+01** (**2.234E+01**) | 1 | 1.681E+02 (7.456E+01) | 5 | 1.762E+02 (6.465E+01) | 8 | 1.564E+02 (3.656E+01) | 4 | 1.724E+02 (3.773E+01) | 6 | 1.340E+02 (3.851E+01) | 3 |
| Avg Rank | 7.00 | | 2.18 | | 1.91 | | 6.18 | | 6.73 | | 4.55 | | 5.55 | | 1.91 (↑ **52%**) | |

configurations of comparison algorithms follow that proposed in corresponding original papers. Experiments are run on Intel i9-10980XE CPU, RTX 3090 GPU and 32GB RAM. When testing, each algorithm executes 10 independent runs and reports the statistical results.

## 5.2 Comparison Analysis

Table 4: Numerical comparison results for DE algorithms on 30$D$ problems, where the mean, standard deviations and performance ranks are reported (with the best mean value on each problem highlighted in **bold**).

| Type | Static | | Adaptive | | | | DRL | | | | | | | | | |
|---|---|---|---|---|---|---|---|---|---|---|---|---|---|---|---|---|
| Algorithm | DE | | MadDE | | NL-SHADE-LBC | | DE-DDQN | | DE-DQN | | LDE | | RLHPSDE | | GLEET-DE | |
| Metrics | Mean (Std) | Rank | Mean (Std) | Rank | Mean (Std) | Rank | Mean (Std) | Rank | Mean (Std) | Rank | Mean (Std) | Rank | Mean (Std) | Rank | Mean (Std) | Rank |
| $f_1$ | 1.658E+09 (5.924E+08) | 8 | 7.098E+03 (7.247E+03) | 3 | 6.234E+03 (6.435E+03) | 2 | 5.362E+08 (4.631E+07) | 6 | 5.659E+08 (4.355E+07) | 7 | 4.563E+04 (3.624E+03) | 4 | 4.863E+04 (3.731E+03) | 5 | **1.639E-01** (**4.818E-01**) | 1 |
| $f_2$ | 6.033E+03 (3.499E+02) | 8 | 2.421E+03 (2.690E+02) | 2 | **2.134E+03** (**2.121E+02**) | 1 | 4.563E+03 (3.612E+02) | 4 | 4.327E+03 (3.121E+02) | 5 | 5.325E+03 (3.205E+02) | 6 | 5.877E+03 (3.643E+02) | 7 | 3.083E+03 (2.838E+02) | 3 |
| $f_3$ | 1.701E+02 (2.740E+01) | 7 | 7.796E+01 (7.228E+00) | 3 | 7.345E+01 (7.048E+00) | 2 | 1.673E+02 (2.445E+01) | 5 | 1.995E+02 (2.853E+01) | 8 | 1.456E+02 (1.656E+01) | 4 | 1.683E+02 (1.343E+02) | 6 | **6.733E+01** (**8.379E+00**) | 1 |
| $f_4$ | 1.741E+01 (8.485E+00) | 8 | **6.168E+00** (**1.003E+00**) | 1 | 6.867E+00 (1.353E+00) | 2 | 1.447E+01 (7.456E+00) | 6 | 1.698E+01 (8.334E+00) | 7 | 8.366E+00 (5.546E+00) | 4 | 1.135E+01 (9.312E+00) | 5 | 7.468E+00 (3.679E+00) | 3 |
| $f_5$ | 4.503E+04 (4.176E+04) | 8 | 1.183E+03 (3.542E+02) | 2 | **1.023E+03** (**3.334E+02**) | 1 | 1.463E+04 (3.642E+03) | 7 | 1.386E+04 (4.545E+03) | 6 | 3.945E+03 (5.645E+02) | 4 | 4.895E+03 (5.997E+02) | 5 | 3.099E+03 (4.503E+02) | 3 |
| $f_6$ | 2.192E+02 (1.003E+02) | 5 | 2.127E+02 (7.915E+01) | 2 | 2.169E+02 (7.043E+01) | 4 | 2.362E+02 (1.406E+02) | 7 | 2.458E+02 (1.556E+02) | 8 | 2.155E+02 (9.615E+01) | 3 | 2.301E+02 (1.137E+02) | 6 | **1.705E+02** (**5.046E+01**) | 1 |
| $f_7$ | 1.056E+04 (9.032E+03) | 8 | 5.156E+02 (2.473E+02) | 3 | 4.675E+02 (2.122E+02) | 2 | 1.037E+03 (6.373E+02) | 6 | 1.652E+03 (5.193E+02) | 7 | 7.893E+02 (2.435E+02) | 4 | 8.961E+02 (3.154E+02) | 5 | **3.788E+02** (**2.277E+02**) | 1 |
| $f_8$ | 1.620E+03 (1.188E+03) | 8 | 1.501E+02 (8.258E+01) | 3 | 1.443E+02 (7.345E+01) | 2 | 7.693E+02 (3.543E+02) | 6 | 7.798E+02 (3.523E+02) | 7 | 3.453E+02 (8.816E+01) | 5 | 3.410E+02 (8.311E+01) | 4 | **1.375E+02** (**7.646E+01**) | 1 |
| $f_9$ | 4.227E+02 (2.089E+01) | 4 | **3.989E+02** (**1.330E+01**) | 1 | 4.231E+02 (1.437E+01) | 5 | 4.313E+02 (2.345E+01) | 7 | 5.015E+02 (2.577E+02) | 8 | 4.132E+02 (1.346E+02) | 3 | 4.303E+02 (1.825E+02) | 6 | 4.105E+02 (1.523E+02) | 2 |
| $f_{10}$ | 5.177E+02 (7.687E+01) | 8 | 2.660E+02 (3.307E+01) | 2 | **2.101E+02** (**2.744E+00**) | 1 | 4.637E+02 (6.756E+01) | 6 | 4.879E+02 (7.231E+01) | 7 | 4.025E+02 (3.445E+01) | 4 | 4.333E+02 (4.131E+01) | 5 | 3.532E+02 (2.357E+01) | 3 |
| $f_{mix}$ | 7.324E+07 (7.104E+07) | 8 | 7.193E+03 (2.803E+03) | 3 | 6.831E+03 (2.435E+03) | 2 | 4.337E+07 (5.156E+07) | 6 | 5.237E+07 (3.460E+07) | 7 | 4.354E+04 (3.641E+03) | 4 | 5.610E+04 (3.451E+03) | 5 | **6.356E+03** (**4.325E+03**) | 1 |
| Avg Rank | 7.27 | | 2.27 | | 2.18 | | 6.00 | | 7.00 | | 4.09 | | 5.36 | | 1.82 (↑ **64%**) | |

### 5.2.1 Comparison among the PSO variants

Table 1 and Table 2 shows the optimization results and the ranks obtained by different PSO algorithms over the 10$D$ and 30$D$ problems respectively. We also present the average performance improvement of GLEET compared with its backbone algorithm (e.g.,GLEET-PSO improves PSO with a **35%** performance gap), which lies on the right of the rank of the GLEET-PSO/GLEET-DMSPSO. It can be observed that the performance of the proposed GLEET-DMSPSO generally and consistently dominates the competitors on both 10$D$ and 30$D$ settings. Both GLEET-PSO and GLEET-DMSPSO significantly improve their back-bones, i.e., PSO and DMSPSO, respectively, which validates the effectiveness of our GLEET in learning generalizable knowledge to control the exploration and exploitation behavior for the algorithms under a given problem class. The superiority of GLEET over the traditional adaptive algorithms sDMSPSO and

Table 5: Generalization experiment results for different problem classes. The "Gap" column shows the difference in optimization performance compared to the original agent (with negative values indicating improvement).

| | | Ag-$f_2$ | | Ag-$f_3$ | | Ag-$f_4$ | | Ag-$f_{mix}$ | | PSO | |
|---|---|---|---|---|---|---|---|---|---|---|---|
| | | Mean (Std) | Gap | Mean (Std) | Gap | Mean (Std) | Gap | Mean (Std) | Gap | Mean (Std) | Gap |
| Simple | $f_2$ | 5.105E+02 (1.776E+02) | 0.000% | 5.164E+02 (1.812E+02) | 1.156% | 5.195E+02 (1.880E+02) | 1.763% | 5.179E+02 (1.871E+02) | 1.450% | 8.428E+02 (2.716E+02) | 65.093% |
| | $f_3$ | 2.213E+01 (5.090E+00) | 4.387% | 2.120E+01 (4.705E+00) | 0.000% | 2.179E+01 (4.786E+00) | 2.783% | 2.175E+01 (4.805E+00) | 2.594% | 2.934E+01 (7.850E+00) | 38.396% |
| | $f_4$ | 1.546E+00 (9.959E-01) | 8.720% | 1.412E+00 (8.379E-01) | −0.703% | 1.422E+00 (7.776E-01) | 0.000% | 1.451E+00 (8.399E-01) | 2.039% | 2.099E+00 (1.235E-01) | 47.609% |
| Complex | $f_5$ | 1.928E+03 (2.374E+03) | 4.356% | 1.592E+03 (1.980E+03) | −13.793% | 1.514E+03 (1.600E+03) | −18.043% | 1.709E+03 (2.126E+03) | −7.482% | 3.395E+03 (5.793E+03) | 83.812% |
| | $f_8$ | 1.480E+02 (9.429E+01) | 2.697% | 1.479E+02 (9.514E+01) | 2.596% | 1.458E+02 (9.395E+01) | 1.194% | 1.447E+02 (9.449E+01) | 0.400% | 3.376E+02 (2.902E+01) | 208.029% |

GLPSO further shows the powerfulness of using a learning-based agent to derive policies instead of manually designed heuristic.

Our GLEET achieves the state-of-the-art performance among learning-based competitors on all test cases when considering the comparison among GLEET-PSO, DRL-PSO and RLEPSO. The three algorithms are all based on DRL, amongst GLEET-PSO and DRL-PSO adopt the same backbone and the RLEPSO improves the EPSO algorithm. Different from the three peer algorithms, we explicitly embed EET information into the state representation to make it more expressive and design an attention-based architecture to make the individuals in population *fully informed* and *exploration-exploitation aware*. With the increasing of dimensions, difficulty of searching surges due to the exponential growth of search space. Facing that difficulty, DRL-PSO and RLEPSO suffer from sharp performance decline, which may be caused by the oversimplified network or the defect of EET information in state representation. Adaptive PSO variants sDMSPSO and GLPSO still have stable performance on high-dimensional problems, which proves that manually designed adaptive PBOs are still competing. Manual adaptive variants sDMSPSO can not improve the backbone DMSPSO on $30D$ problems (considering the average rank). However, on some specific functions (i.e., $f_5$ and $f_7$), sDMSPSO dominates others. This may reveal that adaptive control of EET by manual design has poor generalization. GLEET dominates the optimization performance on composition problem sets ($f_8$, $f_9$ and $f_{10}$) either under $10D$ or $30D$ setting, which indicates that GLEET may perform more stable on complex problem, which is favorable.

#### 5.2.2   Comparison among the DE variants

Table 3 and Table4 shows the optimization results and the ranks obtained by different DE algorithms over the ten problem classes on $10D$ and $30D$ spaces respectively. A consistent conclusion can be deduced as the results above on PSO algorithms. Additionally, it can be noticed that, by taking DE as backbone, the GLEET-DE performs generally better than the GLEET-PSO algorithm.

Besides the conclusions above, we propose an especially challenging task to further examine the control of EET in each algorithm, where we mix all ten problem sets up for training and testing ($f_{mix}$). We trained GLEET-PSO and GLEET-DMSPSO and GLEET-DE on $f_{mix}$. The line $f_{mix}$ in Table 1 to Table 4 shows the optimization results in such a mixed dataset of the GLEET and those competitors on both $10D$ and $30D$ settings, which further verifies that GLEET can not only learns well among similar problems but also learns well among different problem classes.

### 5.3   Generalization Analysis

In the above experiments, the agents are trained on a set of problem instances and tested on another set of unseen ones within the same problem class, which in some ways showed the desired generalization ability of our GLEET. We now continue to evaluate the zero-shot generalization of GLEET under more critical conditions in the following.

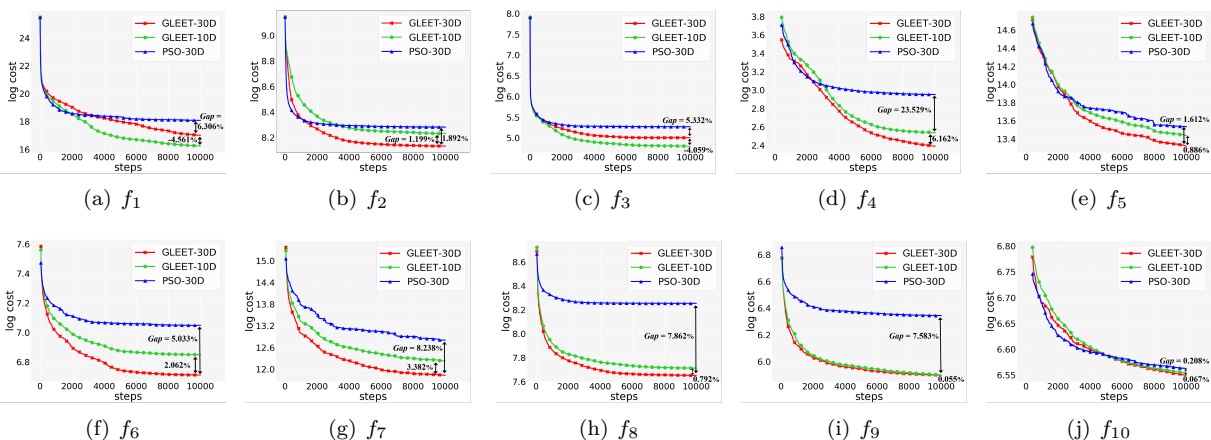

Figure 3: Generalization across different problem dimensions.

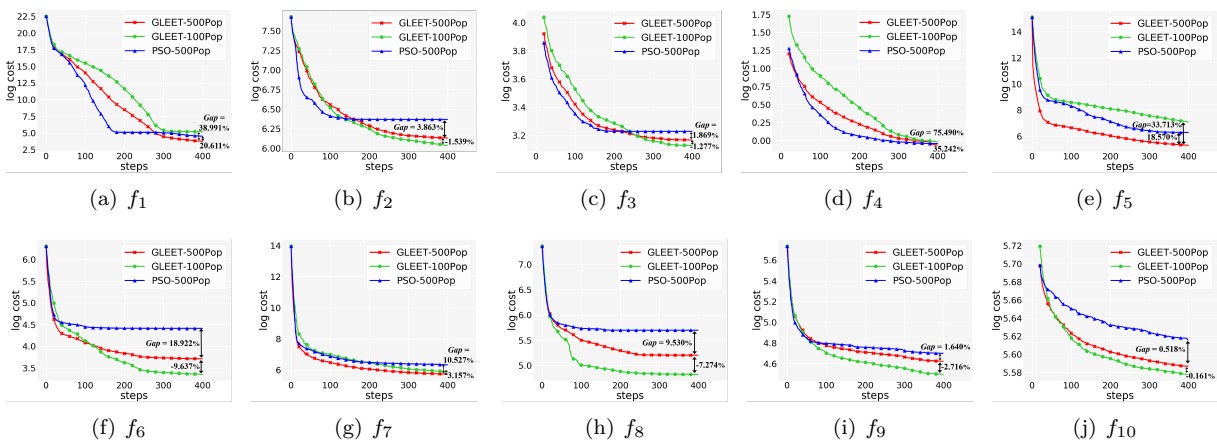

Figure 4: Generalization across different population sizes.

### 5.3.1 Generalization across problems

GLEET is generalizable across problem classes. We train four GLEET-PSO agents, denoted as Ag-$f_2$, Ag-$f_3$, Ag-$f_4$, Ag-$f_{mix}$, on the following four problem sets: the Schwefel ($f_2$) class, the biRastrigin ($f_3$) class, the Grierosen ($f_4$) class and the mixture problems of the above three ($f_{mix}$), and then test their performance on unseen problem classes including more complex problem $f_5$ and $f_8$. Table 5 presents the averaged performance on ten runs and the performance "Gap" between each of the above agent and the original agent trained on the designated problem class. PSO is taken as a baseline and the "Gap" measures its performance difference with GLEET-PSO trained on the designated problem class. The "Gap" is calculated as $\frac{f'-f}{f}$, where $f'$ is the performance of the agent trained on another problem class, and $f$ is performance of the the original agent trained on the designated problem class. "Gap" indicates how much better (less than 0) or worse (greater than 0) when applying a model trained on another problem class. For example, Ag-$f_2$ is trained on $f_2$, when we zero-shot it directly to $f_5$, it achieves a slightly lower performance than the $f_5$'s original agent by 4.356%, which is acceptable. Generally speaking, it can be observed from the table that the agents exhibit promising and even better performance on those different problem classes, which validates the good generalization of the trained policies on unseen problem classes.

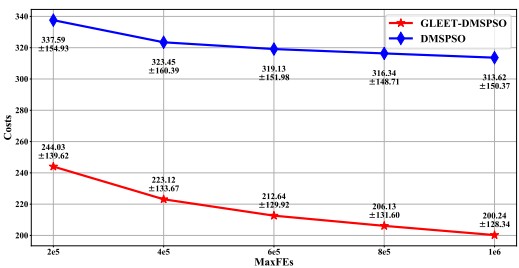
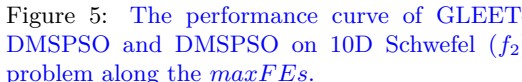

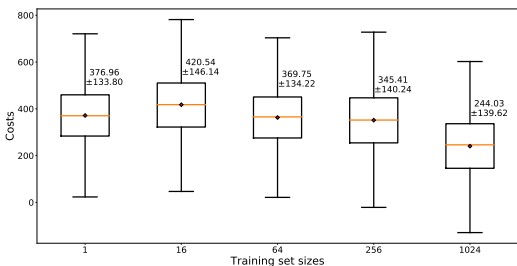

Figure 5: The performance curve of GLEET-DMSPSO and DMSPSO on 10D Schwefel ($f_2$) problem along the $maxFEs$.

Figure 6: The performance curve of GLEET-DMSPSO on 10D Schwefel ($f_2$) problem along the training set size.

### 5.3.2 Generalization across dimensions and population sizes

We train GLEET-PSO on $10D$ problem, denoted as GLEET-$10D$, and apply the model to optimize $30D$ problems. The performance of GLEET-$10D$ is compared with the original PSO and the GLEET-$30D$ model trained on $30D$ problems. Fig. 3 depicts the convergence curves on the 10D problems for illustration, where we also annotate the "Gap" to quantify the final generalization bias. Without any further tuning, GLEET-$10D$ outperforms the original PSO and with a well acceptable gap to GLEET-$30D$ on most of the problems, which further verifies that GLEET is generalizable between different problem dimensions owing to our dimension-free state representation in Section 4.1. In Fig. 4, GLEET-500Pop is the model trained with population size 500 while GLEET-100Pop is trained on population size 100. They are both tested on the scenario of 500 population size. We show that GLEET-100Pop outperforms not only the original PSO but also the GLEET-500Pop on most of the problems. This indicates that, on the one hand, the proposed attention-based en/decoder supports the generalization to a different population size; and on the other hand, through good EET control, a population of 100 particles is sufficient to provide good optimization results.

### 5.3.3 Generalization beyond optimization horizon

Running the agent for more generations than the training horizon is a well-know challenge for meta-learned optimizers. To reveal GLEET's generalization ability across generations, we run GLEET-DMSPSO trained with $2e5$ $maxFEs$ for more generations (up to $1e6$ $maxFEs$) and compare it with DMSPSO. Fig. 5 shows the generalization performance on 10D Schwefel ($f_2$) problems. It can be seen that comparing to the DMSPSO, GLEET-DMSPSO has a larger cost decrease alone the generations, indicating that GLEET agent has learned how to deal with a longer episode and improves the performance of the backbone DMSPSO.

### 5.3.4 Impact of Training set size

Existing RL-based optimizers were trained on a single or a few of problem instances. Although experiments in Section 5.2 have validated the effective of GLEET, the relationship between the performance and the training set size remains to be explore. To showcase the benefits of training the policy on a distribution of problems, we train GLEET-DMSPSO with different training set sizes: 1, 16, 64, 256 and 1024. Taking 10D Schwefel ($f_2$) as a case, the performance of GLEET-DMSPSO is shown in Fig. 6. The results demonstrate that training agents on a set of problem instances may lead to a better performance than training on a single instance, which aligns with our motivations and conclusions. This may because larger training set allows the agent to capture full knowledge about the problem distribution and utilize the knowledge to adaptively control the exploration-exploitation tradeoff which promotes the optimization performance (as done in our GLEET), while training on single instances may lead to overfitting and lose the generalization on unseen instances even in the similar distribution (as done in most existing works).

### 5.3.5 Out-of-distribution Generalization

The generalization analysis in above experiments is conducted within CEC problems. To further evaluate the generalization of GLEET, we introduce a realistic continuous optimization benchmark, Protein-

Table 6: Numerical comparison results on 12D Protein-Docking problems, where the mean, standard deviations, runtime and performance ranks (according to the mean costs) are reported (with the best mean value highlighted in **bold**).

| Type | Static PSO | Adaptive PSO | | | DRL-based PSO | | | |
|---|---|---|---|---|---|---|---|---|
| Algorithm | PSO | DMSPSO | sDMSPSO | GLPSO | DRL-PSO | RLEPSO | GLEET-PSO | GLEET-DMSPSO |
| Metrics | Mean (Std) | Mean (Std) | Mean (Std) | Mean (Std) | Mean (Std) | Mean (Std) | Mean (Std) | Mean (Std) |
| Protein Docking | 5.153E+02 (2.151E+01) | 4.771E+02 (3.345E+01) | 5.101E+02 (3.155E+01) | 5.061E+02 (3.700E+01) | 5.180E+02 (4.940E+01) | 4.895E+02 (4.129E+01) | 4.832E+02 (3.865E+01) | **4.681E+02** **(3.650E+01)** |
| Runtime(s) | 0.539 | 1.156 | 1.137 | 1.103 | 14.341 | 1.307 | 2.134 | 2.515 |
| Rank | 18 | 5 | 16 | 14 | 19 | 11 | 9 | 1 |

| Type | Static DE | Adaptive DE | | DRL-based DE | | | | |
|---|---|---|---|---|---|---|---|---|
| Algorithm | DE | MadDE | NL-SHADE-LBC | DE-DDQN | DE-DQN | LDE | RLHPSDE | GLEET-DE |
| Metrics | Mean (Std) | Mean (Std) | Mean (Std) | Mean (Std) | Mean (Std) | Mean (Std) | Mean (Std) | Mean (Std) |
| Protein Docking | 4.853E+02 (4.331E+01) | 4.803E+02 (4.278E+01) | 4.786E+02 (2.462E+01) | 5.039E+02 (5.681E+01) | 5.211E+02 (6.375E+01) | 4.970E+02 (2.767E+01) | 5.069E+02 (5.082E+01) | 4.794E+02 (2.697E+01) |
| Runtime(s) | 0.943 | 1.565 | 1.486 | 20.687 | 1.734 | 1.791 | 1.895 | 2.467 |
| Rank | 10 | 8 | 6 | 13 | 20 | 12 | 15 | 7 |

| Type | SMAC3 | CMA-ES | | |
|---|---|---|---|---|
| Algorithm | SMAC3-DMSPSO | CMA-ES | IPOP-CMA-ES | PSA-CMA-ES |
| Metrics | Mean (Std) | Mean (Std) | Mean (Std) | Mean (Std) |
| Protein Docking | 5.134E+02 (3.406E+01) | 4.743E+02 (5.972E+01) | 4.716E+02 (4.634E+01) | 4.720E+02 (4.943E+01) |
| Runtime(s) | 1.150 | 4.834 | 23.451 | 15.672 |
| Rank | 17 | 4 | 2 | 3 |

Docking (Hwang et al., 2010). The benchmark contains 280 instances of different protein-protein complexes. These problems are characterized by rugged objective landscapes and are computationally expensive to evaluate. The distribution of Protein-Docking problems is significantly different from the CEC problems, we zero-shot the models agents trained on 10D $f_{mix}$ problems to the Protein-Docking benchmark to evaluate the out-of-distribution generalization of GLEET. Besides, in realistic optimization there are lots of widely adopted sota methods beyond DE and PSO, such as quasi-hyperparameter-free approach CMA-ES (Hansen et al., 2003) and hyper-parameter optimization method SMAC3 (Lindauer et al., 2022). Therefore SMAC3, CMA-ES and its advanced variant IPOP-CMA-ES (Auger & Hansen, 2005), PSA-CMA-ES (Nishida & Akimoto, 2018) with population size adaption are adopted as baselines in the experiment. For SMAC3, we train it on the 10D $f_{mix}$ training problem set to tune the parameter in DMSPSO as GLEET-DMSPSO does, a difference is that the parameter is the same across individuals, iterations and problem instances. Then we apply the learned parameter value to DMSPSO and test it on the Protein-Docking benchmark. All algorithms optimize 1000 $maxFEs$ on each problem instances. We collect their mean values, standard deviations and average runtime for runs in Table 6. Results show that GLEET-DMSPSO outperforms the comparison algorithms. The performance of GLEET indicates that GLEET achieve remarkable generalization performance on out-of-distribution problems. The poor performance of SMAC3-DMSPSO may be because that SMAC3 fixed the hyper-parameter values for all individuals and iterations, which is less flexible for particles to exploration and exploitation in the searching space, while DMSPSO itself has effective adaptive rules which ensure a successful searching.

### 5.4 Interpretability

In this section, we take GLEET-PSO as an example to show GLEET's interpretability since the exploration and exploration tradeoff (EET) is explicitly represented by the "$c_1$" and "$c_2$" in PSO's update formula, which facilitates easy interpretation and analysis. Fig. 8 illustrates how the distribution of EET hyper-parameter $c_{1,i}$ [1] changes along the optimization process for two different problems: a relatively simple problem, the Schwefel ($f_2$) and a difficult one, the Hybrid ($f_6$). Here we let X-axis represent the optimization steps, Y-axis represent the distribution of the output actions, and Z-axis represent the current distribution of $c_{1,i}$

---

[1] $c_{1,i}$ is the individual impact coefficient in Eq. (3) but differs for different particles, while the social impact coefficient is $c_{2,i} = 4 - c_{1,i}$.

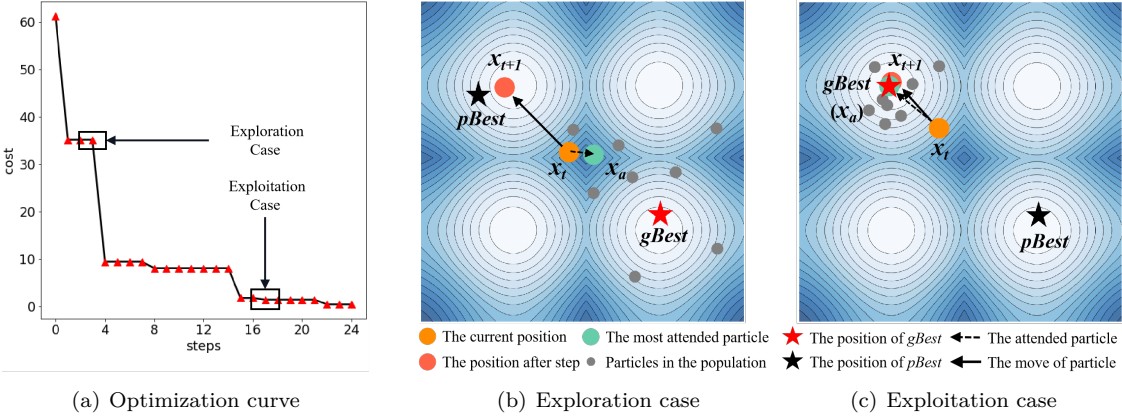

(a) Optimization curve        (b) Exploration case        (c) Exploitation case

Figure 7: Visualization of the attention patterns and the moving of particles during exploration and exploitation controlled by GLEET. In Exploration case, GLEET leans to make the particle as far as possible from the most attended neighbour to get max exploration ability. In Exploitation case, GLEET leans to make the particle as close as possible to the most attended neighbour to reach the global optimum.

for all particles under the control of GLEET. It can be observed that our GLEET automatically controls the EET hyper-parameters throughout the search, with the patterns displayed first in exploration and then in exploitation till the conclusion of the iteration. Meanwhile, it is worth noting that GLEET favors a more complex EET control pattern for difficult problems such as the one shown in Fig. 8(b), where two rounds of exploration and exploitation emerge.

In Fig. 7, we further visualize how the attention among the population (in GLEET-PSO's decoder) affects the moving of particles to perform exploration or exploitation on a 2D toy problem. We highlight an exploration example and an exploitation example in the convergence curve shown in Fig. 7(a). Pertaining to the first case, we observe that the particle made a big improvement after taking the action. In Fig. 7(b), we can see that the particle is located near the centre of the search space, while the majority of the other particles are scattered to the right. The *pBest* and *gBest* particles are in two opposite directions. Note that in this case the most attended particle is a negative sample with a very worse fitness value, the current

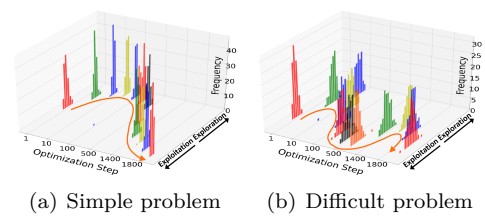

(a) Simple problem      (b) Difficult problem

Figure 8: Visualization of action distribution changes as the optimization process advances.

particle learns to strengthen the utilization of the exemplar located on a much different direction from this negative sample, which hence learns from the *pBest* direction and achieves a great improvement. Pertaining to the second case in Fig. 7(c), the particle attends to the *gBest* and decides to perform exploitation around it. This interpretable behavior verifies our analysis of choosing full attention in Section 4.2.

## 5.5 Ablation study

We first substitute the reward function we have designed in Eq. (10) by other rational ones designed by recently proposed works and observe the performance of these reward functions. We then perform the ablation study on GLEET's own EET embeddings and the attention modules to examine their effectiveness.

### 5.5.1 Analysis on the reward design

Reward quality plays a crucial role in determining the final performance of the policy as it guides the policy update during the training process. In Table 7, we provide a comparison of various practical reward functions recently proposed, including our own approach. Specifically, we consider the reward function $r_1$ proposed

Table 7: The comparison among different reward functions, where the mean and standard deviations of ten runs on the test set are reported (with the best mean value on each problem highlighted in **bold**).

| | Metric | $f_1$ | $f_2$ | $f_3$ | $f_4$ | $f_5$ | $f_6$ | $f_7$ | $f_8$ | $f_9$ | $f_{10}$ |
|---|---|---|---|---|---|---|---|---|---|---|---|
| Ours | Mean | 2.471E+02 | 2.440E+02 | 1.498E+01 | **5.816E-01** | 3.716E+02 | **1.300E+01** | **1.302E+02** | **7.216E+01** | **1.202E+02** | **1.682E+02** |
| | (Std) | (7.676E+02) | (1.396E+02) | (2.357E+00) | **(2.210E-01)** | (1.866E+02) | **(1.020E+01)** | **(8.489E+01)** | **(3.555E+01)** | **(2.016E+01)** | **(1.648E+01)** |
| $r_1$ | Mean | 5.846E+05 | 3.253E+02 | 1.559E+01 | 1.412E+00 | 4.515E+02 | 3.089E+01 | 2.254E+02 | 1.136E+02 | 1.742E+02 | 1.923E+02 |
| | (Std) | (5.353E+05) | (1.433E+02) | (6.938E+00) | (7.098E-01) | (2.043E+02) | (2.492E+01) | (1.577E+02) | (4.948E+01) | (6.542E+01) | (2.184E+01) |
| $r_2$ | Mean | **1.461E+01** | 2.591E+02 | **1.490E+01** | 7.330E-01 | 4.052E+02 | 1.799E+01 | 1.715E+02 | 7.816E+01 | 1.345E+02 | 1.699E+02 |
| | (Std) | **(4.200E+01)** | (1.408E+02) | **(2.272E+00)** | (2.801E-01) | (2.156E+02) | (1.087E+01) | (1.257E+02) | (3.618E+01) | (6.539E+01) | (1.678E+02) |
| $r_3$ | Mean | 4.752E+04 | **2.404E+02** | 1.510E+01 | 6.065E-01 | **3.645E+02** | 2.315E+01 | 2.056E+02 | 8.305E+01 | 1.311E+02 | 1.705E+02 |
| | (Std) | (9.845E+04) | **(1.256E+02)** | (2.283E+00) | (2.045E-01) | **(1.721E+02)** | (1.774E+01) | (2.043E+02) | (3.945E+01) | (6.742E+01) | (1.719E+01) |

Table 8: Ablation studies on the the EET embeddings and the attention modules of GLEET, where the mean and standard deviations of ten runs on the test set are reported (with the best mean value on each problem highlighted in **bold**).

| | Metric | $f_1$ | $f_2$ | $f_3$ | $f_4$ | $f_5$ | $f_6$ | $f_7$ | $f_8$ | $f_9$ | $f_{10}$ |
|---|---|---|---|---|---|---|---|---|---|---|---|
| w/o both | Mean | 3.734E+06 | 6.055E+02 | 2.302E+01 | 1.849E+00 | 2.202E+03 | 9.543E+01 | 7.542E+02 | 3.633E+02 | 2.975E+02 | 2.222E+02 |
| | (Std) | (7.314E+06) | (2.713E+02) | (6.237E+00) | (1.010E+00) | (3.284E+03) | (5.366E+01) | (7.417E+02) | (2.251E+02) | (6.088E+01) | (4.119E+01) |
| w/o EETs | Mean | 3.345E+06 | 5.723E+02 | 2.252E+01 | 1.804E+00 | 2.050E+03 | 7.939E+01 | 7.034E+02 | 3.348E+02 | 2.351E+02 | 2.201E+02 |
| | (Std) | (6.453E+06) | (2.018E+02) | (6.092E+00) | (1.003E+00) | (2.377E+03) | (6.198E+01) | (7.314E+02) | (2.507E+02) | (6.014E+02) | (3.414E+01) |
| w/o MHA | Mean | 3.124E+06 | **5.066E+02** | **2.029E+01** | 1.710E+00 | 2.145E+03 | 8.159E+01 | 6.777E+02 | 3.132E+02 | 2.263E+02 | 2.180E+02 |
| | (Std) | (5.269E+06) | **(1.767E+02)** | **(4.273E+00)** | (1.046E+00) | (2.540E+03) | (5.959E+01) | (6.397E+02) | (2.588E+02) | (6.031E+01) | (3.635E+01) |
| GLEET | Mean | **2.748E+06** | 5.105E+02 | 2.120E+01 | **1.422E+00** | **1.847E+03** | **4.449E+01** | **4.977E+02** | **1.096E+02** | **1.665E+02** | **1.882E+02** |
| | (Std) | **(4.205E+06)** | (1.776E+02) | (4.705E+00) | **(7.776E-01)** | (2.552E+03) | **(3.381E+01)** | **(3.549E+02)** | **(3.924E+01)** | **(6.137E+01)** | **(4.127E+01)** |

by Yin et al. (Yin et al., 2021), which assigns a reward of 1 for improvement and $-1$ otherwise:

$$r_1^{(t)} = \begin{cases} 1 & \text{if} \quad f(gBest^{(t)}) < f(gBest^{(t-1)}) \\ -1 & \text{otherwise} \end{cases} \tag{13}$$

Sun et al. (Sun et al., 2021) introduced another reward function, denoted as $r_2$, which measures the relative improvement between consecutive steps as the reward:

$$r_2^{(t)} = \frac{f(Best^{(t-1)}) - f(Best^{(t)})}{f(Best^{(t-1)})} \tag{14}$$

where $Best^{(t)}$ is the best particle in the $t$ time step population.

Furthermore, Xue et al. (Xue et al., 2022) identified the issue of premature convergence in PBOs and proposed a novel triangle-like reward function, denoted as $r_3$, to address this concern:

$$r_3^{(t)} = (1/2) \cdot (p_{t+1}^2 - p_t^2), \tag{15}$$

$$p_{t+1} = \begin{cases} \frac{f(g^{(0)}) - f(g^{(t)})}{f(g^{(0)})} & \text{if} \quad f(g^{(t)}) < f(g^{(t-1)}) \\ p_t & \text{otherwise} \end{cases} \tag{16}$$

where $g^{(t)}$ denotes $gBest^{(t)}$.

We train GLEET-PSO with these reward functions and compare their optimization results. It turns out that under the experiment setting in this paper, our reward function stands out. In comparison, $r_1$ presents poor performance on all problems, $r_2$ and $r_3$ are acceptable on simpler problems. Notably, our reward function demonstrated better performance on more complex problems. We recognize the importance of investigating this issue further in future work, with the aim of designing more compatible and effective reward functions.

### 5.5.2 Analysis on the network design

We now conduct ablation studies to verify the effectiveness of our state representation (Section 4.1) and network designs (Section 4.2) in the instantiation of GLEET to PSO. Specifically, we compare GLEET-PSO with its degraded versions "GLEET w/o EETs", "GLEET w/o MHA", and "GLEET w/o both". Here, the first version removes the exploration and exploitation features in the states and thus the decoder could only perform self-attention based on FIPEs given that EETs are no longer available; the second variant removes all the MHA in the en/decoders; and the third variant removes both the first and the second designs. The results presented in Table 8 demonstrate the critical importance of both EETs and MHA in GLEET. The inclusion

Table 9: Ablation studies on the the state representation, where the mean and standard deviations of ten runs on the test set are reported (with the best mean value on each problem highlighted in **bold**).

| | Metric | $f_1$ | $f_2$ | $f_3$ | $f_4$ | $f_5$ | $f_6$ | $f_7$ | $f_8$ | $f_9$ | $f_{10}$ |
|---|---|---|---|---|---|---|---|---|---|---|---|
| w/o $s_{1\sim4}$ | Mean | 3.546E+06 | 6.672E+02 | 2.412E+01 | 1.745E+00 | 2.163E+03 | 7.329E+01 | 7.588E+02 | 5.971E+02 | 3.610E+02 | 2.876E+02 |
| | (Std) | (6.675E+06) | (2.965E+02) | (6.153E+00) | (1.331E+00) | (2.974E+03) | (4.993E+01) | (7.464E+02) | (3.699E+02) | (6.631E+01) | (4.273E+01) |
| w/o $s_{5\sim6}$ | Mean | 3.376E+06 | 6.034E+02 | 2.358E+01 | 1.773E+00 | 2.189E+03 | 6.912E+01 | 7.331E+02 | 2.062E+02 | 3.034E+02 | 2.766E+02 |
| | (Std) | (5.975E+06) | (2.311E+02) | (6.274E+00) | (1.231E+00) | (2.112E+03) | (6.134E+01) | (7.762E+02) | (2.371E+02) | (6.237E+02) | (4.130E+01) |
| w/o $s_{7\sim9}$ | Mean | 3.383E+06 | 5.977E+02 | 2.316E+01 | 1.786E+00 | 2.107E+03 | 6.812E+01 | 7.201E+02 | 4.691E+02 | 2.981E+02 | 2.537E+02 |
| | (Std) | (5.668E+06) | (2.232E+02) | (5.985E+00) | (1.187E+00) | (2.313E+03) | (6.900E+01) | (6.861E+02) | (2.743E+02) | (6.217E+01) | (3.935E+01) |
| GLEET | Mean | **2.748E+06** | **5.105E+02** | **2.120E+01** | **1.422E+00** | **1.847E+03** | **4.449E+01** | **4.977E+02** | **1.096E+02** | **1.665E+02** | **1.882E+02** |
| | (Std) | (4.205E+06) | (1.776E+02) | (4.705E+00) | (7.776E-01) | (2.552E+03) | (3.381E+01) | (3.549E+02) | (3.924E+01) | (6.137E+01) | (4.127E+01) |

of EETs allows for explicit incorporation of exploration and exploitation information from the optimization process into the decoders. The fully-informed MHA plays a crucial role in facilitating the learning of more useful features through the interaction between individual embeddings of the population. This finding partly justifies the oversimplification of the network architectures used in existing learning-based approaches.

### 5.5.3 Analysis on the state representation

In this section we conduct ablation studies on the features in state representation. As introduced in Section 4.1.1, there are nine features in the state which can be divided into three parts: the features about search progress ($s_{1\sim4}$), the distribution of costs ($s_{5\sim6}$) and about population distribution ($s_{7\sim9}$). To evaluate their effect we ablate each of them from GLEET-PSO agent and train them on the 10D problems. Their performance is shown in Table 9 where GLEET is the baseline with full state features. Results indicates that firstly removing any one of the three parts features would significantly affect the learning effectiveness of GLEET. Besides, the optimization progress feature $s_1 \sim s_4$ contribute most to GLEET, which can be interpreted as a informative signal telling GLEET when and where to adjust the hyper-parameter values for better searching behaviour. A comprehensive state representation would help learning indeed.

### 5.5.4 Analysis on the RL hyper-parameters

RL algorithms can be very sensitive to hyper-parameters, to make a deeper analysis on GLEET we explore the effect of batch size and trajectory length $T$. The batch size may not influence the training stability. In PPO, the length of trajectory segments could impact the reward accumulation and the later learning steps. For the batch size we compare the performance with 8, 16, 32 ad 64 batch sizes. For the trajectory length we adopt the values of 5, 10 and 20. The other hyper-parameters are frozen in the experiment. Taking GLEET-DMSPSO with 10D Schwefel ($f_2$) as a case, the results for batch sizes and trajectory lengths are shown in Fig. 9 and Fig. 10 respectively. The experimental results show that GLEET-DMSPSO with 8, 16, 32, and 64 batch sizes consistently exhibits good performance on 10D Schwefel ($f_2$) problems regardless of the numbers of the batch size, and the results of varying trajectory lengths reveal that a proper and moderate length may benefit the final performance since a shorter trajectory may increase the training variance and an overlong trajectory may reduce leaning steps in episodes (10 length trajectory has 200 K-epoch learning in a 2000 generation episode but the 20 length one only has 100) which may degrade the performance.

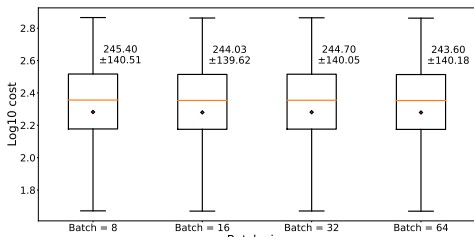

Figure 9: The boxplots of GLEET-DMSPSO on Schwefel ($f_2$) problem with different batch sizes.

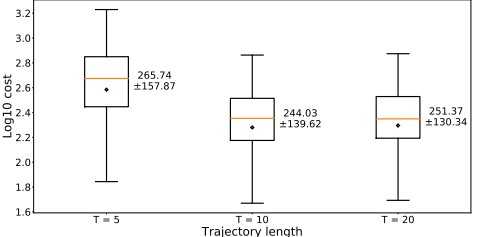

Figure 10: The boxplots of GLEET-DMSPSO on Schwefel ($f_2$) problem with different trajectory length $T$.

## 6    Conclusion

This paper proposed a generalizable GLEET framework for dynamic hyper-parameters tuning of the EET issue in PBO algorithms. A novel MDP was formulated to support training on a class of problems, and then inference on the other unseen ones. We instantiated the GLEET to well-known PBO algorithms by specially designing an attention-based network architecture that consists of a feature embedding module, a fully informed encoder, and an exploration-exploitation decoder. Experimental results verified that GLEET not only improves the backbone algorithms significantly, but also exhibits favorable generalization ability across different problem classes, dimensions, etc. However, there are still some limitations in this study. Although GLEET has shown the state-of-the-art generalization performance among the RL-based methods, the handcrafted population features and EET features based on the Fitness Landscape Analysis may still show vulnerability to high-dimensional scenario. Besides, GLEET may not be able to be applied on backbone optimizers without explicit EET control parameters. Moreover, the population size of the backbone optimizer can not change dynamically which changes the action space of the MDP and makes the training meaningless. Future work includes but is not limited to addressing the above limitations, in order to further boost the performance of learning-based PBOs. Additionally, it deserves to research into a standardized evaluation environment to facilitate the comprehensive comparisons among different learning-based PBO methods.

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

## A    RL Hyper-parameter

Table 10 shows the used Hyper-parameters.

Table 10: The used Hyper-parameters

| | | Parameters | Values |
|---|---|---|---|
| Dataset | | Training set size | 128 |
| | | Testing set size | 1024 |
| | | Batch size | 16 |
| | | Problem dim | 10D, 30D |
| | | $maxFEs$ | $2e5$, $1e6$ |
| | | Search space | $[-100, 100]$ |
| | | Number of runs | 10 |
| PPO | | $MaxEpoch$ | 100 |
| | | Trajectory length $T$ | 10 |
| | | K-epoch $\kappa$ | 3 |
| | | Learning rate $lr$ | $4e-5$ to $1e-5$ |
| | | Gamma $\gamma$ | 0.999 |
| | | Eps clip | 0.1 |
| | | Max gradient norm | 0.1 |
| Network | | Feature input dim $K$ | 9 |
| | | Population Embedding dim | 128 |
| | | Exploration Embedding dim | 128 |
| | | Exploitation Embedding dim | 128 |
| | | EET Embedding dim | 128 |
| | | Attention heads | 8 |
| | | Actor hidden dim | $32 \times 8$ |
| | | Critic hidden dim | $32 \times 16$ |
| | | Controlled dim $M$ (PSO) | 1 |
| | | Controlled dim $M$ (DMSPSO) | 1 |
| | | Controlled dim $M$ (DE) | 3 |
| | | EET Embedding dim | 128 |
| GLEET-PBO | PSO | Population size $N$ | 100 |
| | | Max velocity | 10 |
| | | Inertial factor | 0.728 |
| | DMSPSO | Population size $N$ | 99 |
| | | Sub-swarm size | 3 |
| | | Regroup generation | 10 |
| | | Max velocity | 10 |
| | | Inertial factor | 0.728 |
| | DE | Population size $N$ | 100 |

## B    Boxplots for numerical comparison results

Fig. 11 presents the boxplots of comparison results between PSO methods on 10D (Table 1) and 30D (Table 2) problems. Fig. 12 shows the boxplots of comparison results between DE methods in Table 3 and Table 4. Due to the various scales of costs, the boxplots present the log10 values. It can be seen that although the costs could be largely variant on some problems (e.g., variants on $f_1$ can up to three orders of magnitude), the overall ranking and the conclusions are invariant with that in Section 5.2. GLEET significantly improves the backbone algorithms and dominates the optimization performance on 10D and 30D problems.

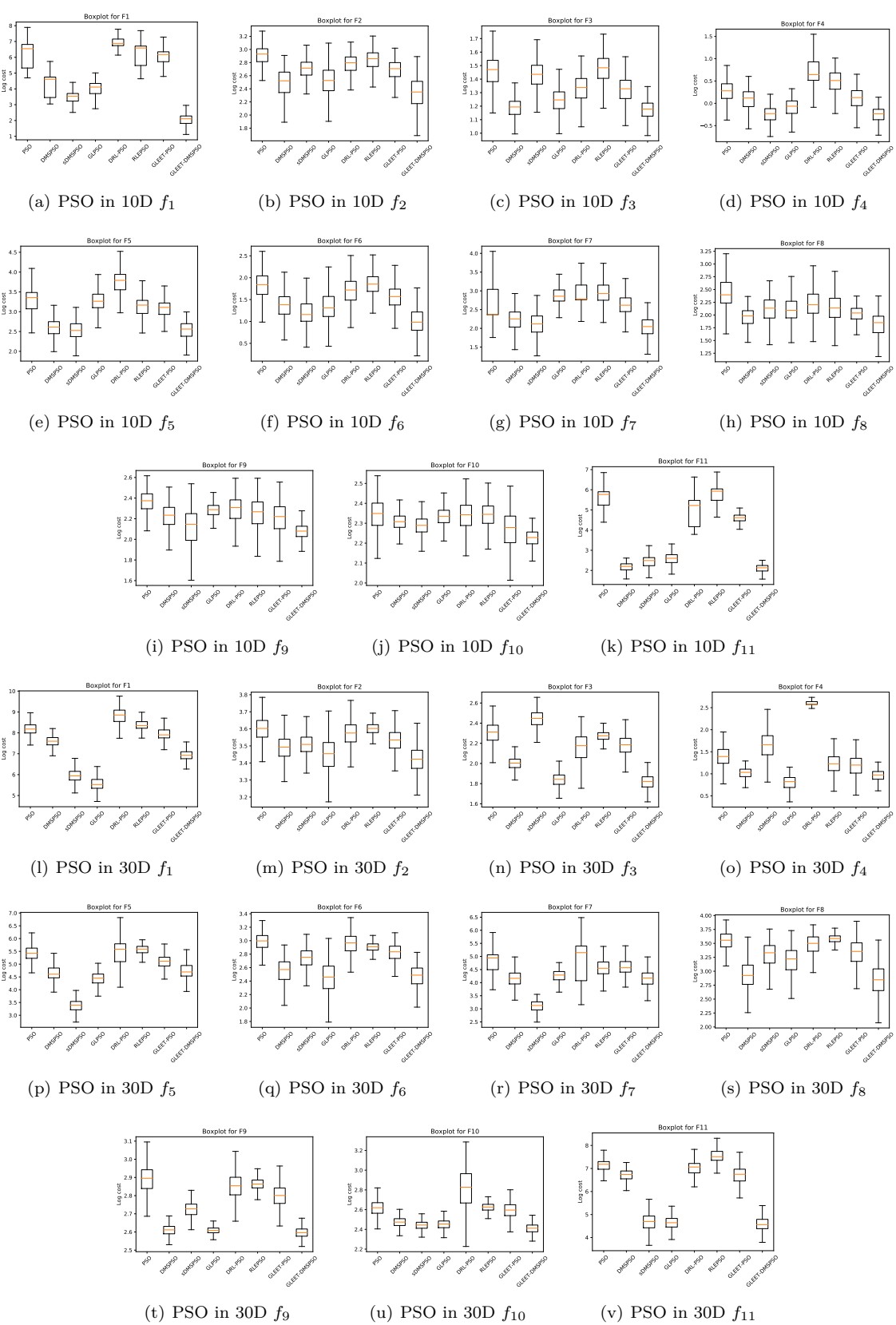

Figure 11: The boxplots for the comparison results of Table 1 and Table 2

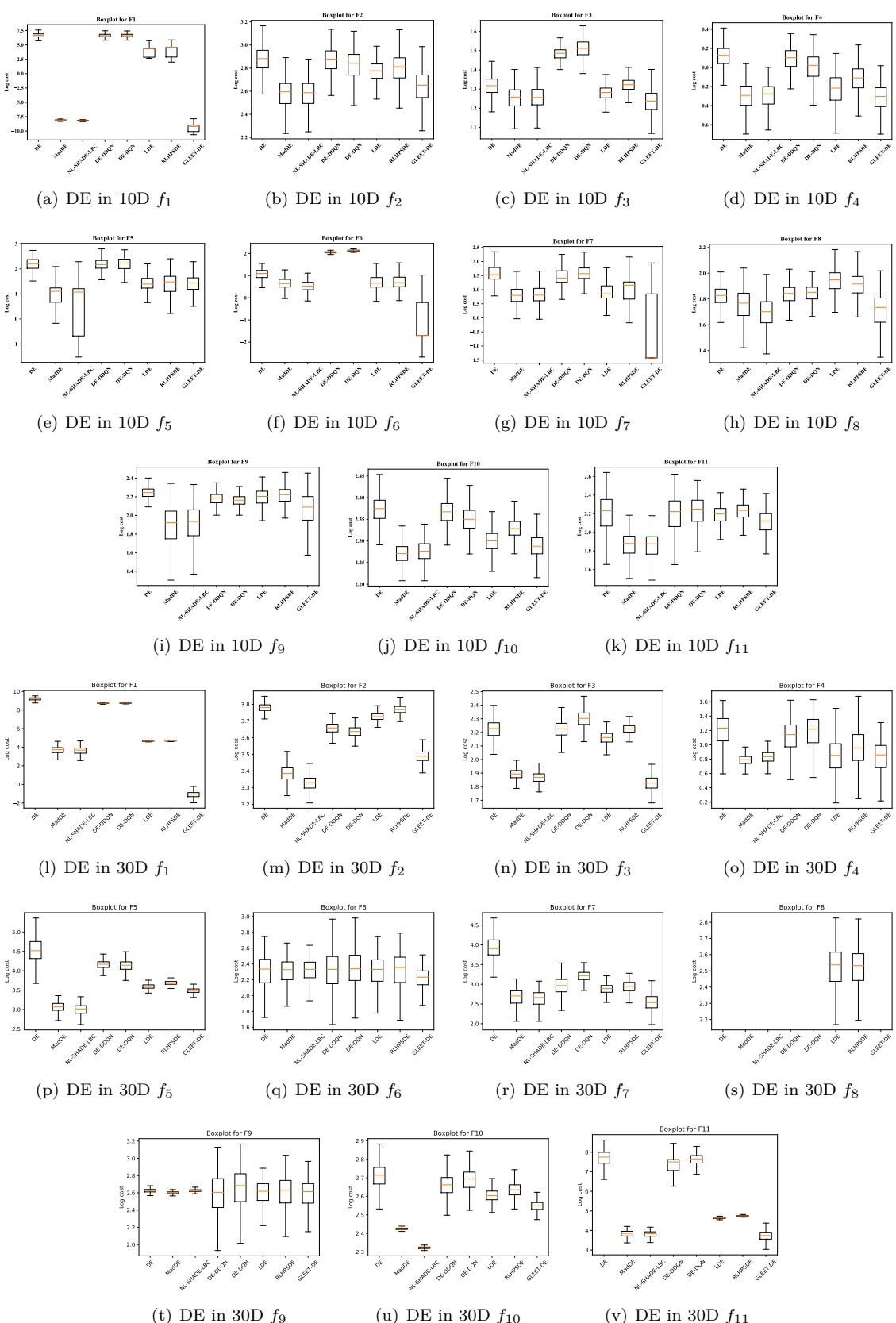

Figure 12: The boxplots for the comparison results of Table 3 and Table 4

