# OpenReview forum: "Auto-configuring Exploration-Exploitation Tradeoff in Population-Based Optimization: A Deep Reinforcement Learning Approach"
_TMLR — Rejected by TMLR_

### Review · Reviewer_gD5U · 2023-08-16

**Summary Of Contributions:**

This paper addresses the exploration-exploitation tradeoff (EET) problem in population based optimization algorithms with a novel approach that leverages a distribution of problems to learn a generalizable policy.

Contributions:
* Apply Deep Reinforcement Learning to a distribution of problems to favor the generalization to unseen instances.
* State representation that contains EET information.
* Transformer-style network architecture to effectively extract and process the EET features.


**Audience:**

Yes

**Broader Impact Concerns:**

I do not have any concern about the implications of this work.

**Claims And Evidence:**

No

**Requested Changes:**

Crucial changes to secure my recommendation for acceptance:
* Box plots (+ the raw data) as the standard deviation is often too large to really distinguish the different algorithms.
* Generalization analysis with a plot that shows the generalization gap in function of the number of problem instances the agent was trained on.
* Add a Table with all the hyperparameters used (including the hyperparameters for the PPO algorithm).

Changes that will strengthen the work:

* Related work on generalization in RL (e.g. [1], [2])


References:

[1]:  Cobbe, Karl, et al. "Leveraging procedural generation to benchmark reinforcement learning." International conference on machine learning. PMLR, 2020.

[2]: Kirk, Robert, et al. "A survey of zero-shot generalisation in deep reinforcement learning." Journal of Artificial Intelligence Research 76 (2023): 201-264.



**Strengths And Weaknesses:**

## Strength:

* Clear presentation of the problem and the method. I particularly appreciated Figure 2 which illustrates the network design.
* Ablation study on the reward function, the state space (with or without the EET features), and the network design (with or without the Multi Head Attention) which covers the main contributions of the paper.
* Extension of the CEC 2021 benchmark to make it possible to train on a large distribution of problems.


## Weaknesses:

* Lack of information on the hyperparameters of the RL algorithm. RL algorithms can be very sensitive to hyperparameters and most of the hyperparameters for PPO are missing which is problematic for the reproducibility. A deeper analysis on that front would be nice. An example of such analysis can be found in the section V of [1] where the authors explain that too much diversity in the number of environments run in parallel could cause instabilities.

* Lack of related work on generalization to new environments in RL. Training on a distribution of problems and generalizing to new ones is a key difference with the previous works that use Deep RL but there is no related work on this subject.

* Equation (11):
$\hat{e}$ and $e^{(l)}$ depend on each other. It looks like an initial condition is missing as the value of $e^{(0)}$ is not explicitly given.

* The tables with the results contain the mean, std, and rank of the scores of each algorithm. However, due to the large standard deviation, the top performing algorithms are often within a standard deviation from one another. Hence, knowing only the mean and the standard deviation is not enough to clearly distinguish the best algorithms. One solution could be to use box plots to have a better understanding of the variance between runs. In addition, sharing the scores for each run would be useful so that the readers can make their own visualizations.

* The generalization analysis:
     * The generalization experiment results for different problem classes that are displayed in Table 5 are hard to interpret as there are no comparisons with baselines.
     * To showcase the benefits of training the policy on a distribution of problems, I would compute the generalization gap in function of the number of problems that were used for training (starting with only one problem which would be a nice baseline to isolate the importance of training on multiple problems). With the results of Table 5, one cannot rule out that the policy can generalize to unseen problem even when trained on a single problem instance.
* In section 5.3.2:
I did not find any definition of GLEET-500Pop so it is not clear what the `generalization` refers to in `the proposed attention-based en/decoder supports the generation to a different population size`. Was GLEET-500Pop trained on 100 particles and tested on 500 particles or the other way around?
* Clarification: are the baselines (e.g. DRL-PSO, RLEPSO) trained on a distribution of problems as well? If yes, were any baselines tuned?

References:

[1]:  Cobbe, Karl, et al. "Leveraging procedural generation to benchmark reinforcement learning." International conference on machine learning. PMLR, 2020.

---

> ### Author Response · Authors · 2023-09-12
> **Response#1 to Reviewer #gD5U**
>
> We sincerely appreciate your thorough review and valuable feedback on our paper. Thank you for acknowledging that our presentation is clear and our ablation studies could support the main contributions. We have carefully addressed each of your comments and provided a point-by-point response below.
>
> > Lack of information on the hyperparameters of the RL algorithm. RL algorithms can be very sensitive to hyperparameters and most of the hyperparameters for PPO are missing which is problematic for the reproducibility. A deeper analysis on that front would be nice. An example of such analysis can be found in the section V of [1] where the authors explain that too much diversity in the number of environments run in parallel could cause instabilities.
> >
>
> > Lack of information on the hyperparameters… A deeper analysis on that front…
> >
>
> Thanks for your valuable suggestion. Following the suggestions, we have conducted a deeper analysis on the effects of hyperparameters. For ease of reference, we have added a summary table of hyperparameters in Appendix A of our revised paper. We then followed the suggestions to conduct analysis on crucial hyper-parameters and added box-plots in Section 5.5.4 of our revised paper, where the effects of batch size and trajectory length are evaluated. The experimental results show that GLEET-DMSPSO with 8, 16, 32, and 64 batch sizes consistently exhibits good performance on 10D Schwefel ($f_2$) problems regardless of the numbers of the batch size, and the results of varying trajectory lengths reveal that a proper and moderate length may benefit the final performance since a shorter trajectory may increase the training variance and an overlong trajectory may reduce leaning steps in episodes (10 length trajectory has 200 K-epoch learning in a 2000 generation episode but the 20 length one only has 100) which may degrade the performance.
>
> > Lack of related work on generalization to new environments in RL. Training on a distribution of problems and generalizing to new ones is a key difference with the previous works that use Deep RL but there is no related work on this subject.
> >
>
> We understand your concern about the related works. However, we would like to clarify that RL-based black-box optimization (RL-BBO) is a field still in its formative stages (it has only been developed for around 5 years). To the best of our knowledge, although there are lots of researchers who notice the importance of training and testing data distributions, existing works in the field of RL-BBO usually train their policies on single instances or a few benchmark problems without the consideration of distribution generalization. The discussions about the lack of generalization issue in this field are explained in the revised introduction section (see the blue parts), which is also considered as one of our motivations to propose GLEET, a pioneering work in the RL-BBO field that promotes better generalization of the trained DRL policy.
>
> > Equation (11): $\hat{e}$ and $e^{(l)}$ depend on each other. It looks like an initial condition is missing as the value of $e^{(0)}$ is not explicitly given.
> >
>
> Sorry for the confusion. The initial condition $e^{(0)}$ is the population embeddings embedded from the population features as shown in Figure 2. We have added the corresponding description to the revised paper (colored in blue).
>
> > The tables with the results contain the mean, std, and rank of the scores of each algorithm. However, due to the large standard deviation, the top performing algorithms are often within a standard deviation from one another. Hence, knowing only the mean and the standard deviation is not enough to clearly distinguish the best algorithms. One solution could be to use box plots to have a better understanding of the variance between runs. In addition, sharing the scores for each run would be useful so that the readers can make their own visualizations.
> >
>
> Thanks for your suggestion. In the revised Appendix B, we present the boxplots of numerical comparison results in Tables 1 to 4. It can be seen that although the standard deviation of costs could be largely variant on some problems, the overall ranking and the conclusions remain the same as that in Section 5. GLEET consistently significantly improves the backbone algorithms and dominates the optimization performance on 10D and 30D problems. To facilitate a more detailed comparison and understanding of the performance, we will release the raw comparison data in the supplementary material containing 128 instances for each problem and each algorithm. Readers can access the scores and make their own visualizations for better understanding.

---

> > ### Author Response · Authors · 2023-09-12
> > **Response#2 to Reviewer #gD5U**
> >
> > > The generalization experiment results for different problem classes that are displayed in Table 5 are hard to interpret as there are no comparisons with baselines.
> > >
> >
> > Thanks for your suggestion. We have added PSO as the baseline in Table 5. It can be seen that all GLEET agents, including those original and generalized, outperform PSO significantly.
> >
> > > To showcase the benefits of training the policy on a distribution of problems, I would compute the generalization gap in function of the number of problems that were used for training (starting with only one problem which would be a nice baseline to isolate the importance of training on multiple problems). With the results of Table 5, one cannot rule out that the policy can generalize to unseen problem even when trained on a single problem instance.
> > >
> >
> > Thanks for your valuable suggestion. We have added such baselines in the revised Section 5.3.4. Following your suggestions, we compare the generalization performance of the policies trained with different training set sizes (i.e., 1, 16, 64, 256 and 1024 instances). To ensure fairness, we vary their *MaxEpoch* to retain the same learning steps for gradient updates. The results shown in Figure 6 demonstrate that training agents on a set of problem instances may lead to a better performance than training on a single instance, which aligns with our motivations and conclusions. This may be because a larger training set allows the agent to capture full knowledge about the problem distribution and utilize the knowledge to adaptively control the exploration-exploitation tradeoff which promotes the optimization performance (as done in our GLEET), while training on single instances may lead to overfitting and degrade the generalization on unseen instances even in a similar distribution (as done in most existing works).
> >
> > > In section 5.3.2: I did not find any definition of GLEET-500Pop so it is not clear what the `generalization` refers to in `the proposed attention-based en/decoder supports the generation to a different population size`. Was GLEET-500Pop trained on 100 particles and tested on 500 particles or the other way around?
> > >
> >
> > We would like to clarify that GLEET-500Pop is the model trained with a population of size 500 while GLEET-100Pop is trained on a population of size 100. They are both tested on the scenario of 500 population size. We have refined the corresponding descriptions in the revised paper (colored in blue).
> >
> > > Clarification: are the baselines (e.g. DRL-PSO, RLEPSO) trained on a distribution of problems as well? If yes, were any baselines tuned?
> > >
> >
> > Yes. For the learning-based baselines, we train them on the same training dataset (a distribution of problems) as GLEET. To do so, we made some essential modifications to allow them to conduct training on a given distribution of problems. Specifically, baselines including DRL-PSO, DE-DDQN, DE-DQN, and RLHPSDE are trained and tested on a single problem instance in their original references. To adapt them to our training set, we set their batch size to 16 (the same as GLEET) and averaged the batch to compute the loss (the same as GLEET). Meanwhile, we ensure that the learning steps of all learning-based baselines are set to be the same as GLEET to ensure fairness. We follow their recommended setups for other hyperparameters.

---

### Review · Reviewer_SVNG · 2023-08-21

**Summary Of Contributions:**

This paper proposes a reinforcement learning based online hyper-parameter selecting scheme for population-based black-box optimization algorithms such as PSO and DE.

In the proposed framework, the hyper-parameters for each individual in the population are generated by the policy network, where the state is manually-designed metric that reflect the progress of the individual, etc., and the action is the proposed hyper-parameter values for each individual. The policy is pre-trained on a set of training problem instances that are expected to be similar to the test problem. Differently from other hyper-parameters tuning schemes that requires searching for the best hyper-parameter value on the fly, the proposed approach do not require any fine-tuning on the test problem itself.

The proposed approach has been compared to the static variant, where the hyper-parameter values are pre-defined, some adaptive variant that searches adequate hyper-parameter values on the fly, and some learning-based approaches that train a policy controlling the hyper-parameter values on training problem instances.

**Audience:**

Yes

**Broader Impact Concerns:**

None.

**Claims And Evidence:**

No

**Requested Changes:**

Please address the above mentioned limitations.

Here are additional comments:

The authors claim that the oversimplified state representation and network architecture are the cause of the inefficacy of the existing learning-based approaches in the introduction. However, no empirical or theoretical evidence has been provided in the paper. Please provide some evidence, possibly by adding some ablation study.
It is not clear what is meant by “generalization” (e.g., in Section 1) in this context as the test and training problem instances must be similar. Please clarify this point.

**Strengths And Weaknesses:**

Strength: transformer-based hyper-parameter selection scheme trained by reinforcement learning approaches on prepared training problem instances.

Limitations 1. Comparison.

One of the possible advantages of the proposed approach over other HPO schemes such as SMAC is that the hyper-parameter values are possibly different for individuals and for iterations. However, this point was not evaluated in the experiments in the paper. Moreover, the proposed approach has not been compared with such HPO schemes. Because these HPO schemes (such as SMAC) requires training problem instances and do not require any optimization on the test problem, the setting is the most relevant to this paper. The proposed approach should be compared to these approaches to show the usefulness of the proposed approach. Otherwise, the other comparisons done in the paper is not quite fair as only the proposed approach variants perform pre-training on a training problem instances. Moreover, for adaptive variants, one could run these variants on training problems, then transfer the results to the test problem, possibly as the initial hyper-parameter values and to limit the range of the hyper-parameter values. In this way, the comparison between adaptive variants and learning-based variants becomes fairer.

Limitations 2. Experimental Setting.

This is the most critical point. The authors create training problem instances and test problem instances from the SAME function by applying different translation and rotation. The test and training instances are too similar and such a similar training problem instance is hardly prepared in practice as the target problem is black-box. It is not at all surprising that the good hyper-parameter values are the same for the test and training problem instances as they are isometric. The authors must use the different set of problems as the test and the training problem instances. Use CEC testbed as training, and some real-world applications as testing, for example.

Limitation 3. Baseline.

As baseline approaches, the authors employ PSO and DE, both of which requires parameter tuning depending on the problem instances. Therefore, the hyper-parameter tuning scheme is required. However, a very natural question arises as how much the proposed approach gains over quasi-hyperparameter-free approaches such as IPOP-CMA-ES and PSA-CMA-ES at the cost of possibly non-negligible training time? The time (and the number of function evaluations) for the training must be reported and discuss whether the performance gain over the baselines and other SOTA approaches such as CMA-ES are significant for spending the training time. Please include quasi-hyperparameter-free approaches such as IPOP-CMA-ES and PSA-CMA-ES as the comparison baseline to show the usefulness of the proposed approach.

Limitation 4. Clarity.

Differences from the existing approaches are not clearly stated in the related works section. The authors commented in the introduction “Existing works stipulated DRL to be conducted online, where they trained and tested the model directly on the target problem instance, that is, their methods require (re-)training for every single problem.” In this sentence “existing works” includes all the paper cited in Section 2? Does it include DRL-PSO and RLEPSO? The statements should be clarified.

Limitation 5. Generalization.

The authors investigated the generalization ability of the proposed approach in Section 5.3. However, the benchmark problems used in this investigations are all multimodal problems. Therefore, they are all similar. For such problems, I suspect that adequate hyper-parameter values are all similar. Therefore, not only the proposed approach but also existing approaches could perform well. Because the authors claim in the introduction that the generalization ability is the issue of the existing approaches, comparison with existing approaches must be included in this experiments. Moreover, to evaluate generalization ability, one should construct a bit more different sets of test and training scheme.

---

> ### Author Response · Authors · 2023-09-12
> **Response#1 to Reviewer #SVNG**
>
> We sincerely appreciate your thorough review and valuable feedback on our paper. We have carefully followed your suggestions, added a zero-shot generalization study on the realistic Protein-Docking benchmark, and thoroughly revised our paper with other new results. We hope our point-by-point response and new results below could clear your concerns about our paper.
>
> > Limitations 1. Comparison.
> >
> >
> > One of the possible advantages of the proposed approach over other HPO schemes such as SMAC is that the hyper-parameter values are possibly different for individuals and for iterations. However, this point was not evaluated in the experiments in the paper. Moreover, the proposed approach has not been compared with such HPO schemes. Because these HPO schemes (such as SMAC) requires training problem instances and do not require any optimization on the test problem, the setting is the most relevant to this paper. The proposed approach should be compared to these approaches to show the usefulness of the proposed approach.
> >
>
> We thank the reviewer for pointing out HPO schemes. The HPO schemes such as SMAC3 (https://arxiv.org/abs/2109.09831) can be viewed as an integrated framework leveraging Bayesian Optimization for tuning Machine Learning algorithms (e.g., SVM, XGboost, etc). We agree with the reviewer that they could be regarded as relevant to our paper in a general view though we have something different from HPO schemes such as the surrogate model used (Gaussian Process Regression vs neural network). Following the suggestions and given that BO methods are good at solving expensive evaluation problems, we then compare our GLEET with the SMAC3 on an additional expensive-to-evaluate test set from Protein-Protein Docking benchmark (https://onlinelibrary.wiley.com/doi/abs/10.1002/prot.22830), which include $280$ protein docking tasks with extremely rugged searching surfaces and large searching spaces $[-inf,inf]^{12}$. Specifically, we train SMAC3 and our GLEET for optimizing hyper-parameter values of a PSO variant (DMSPSO) on the $f_{mix}$ test set (including all $10$ problem types in CEC benchmark, 128 samples with different rotation and shift). After training, we test the original DMSPSO, DMSPSO tuned by SMAC3, and the trained GLEET-DMSPSO on the protein docking test set for $10$ independent runs. The optimization results are highlighted in the Tabe below, where we can conclude that:
>
> - Although HPO schemes such as SMAC3 show similar ambition with GLEET, GLEET may gain more generalization performance through learning a neural network, while SMAC3 is limited by its Gaussian model.
> - The DMSPSO tuned by SMAC3 performs even worse than the original DMSPSO, this may be because SMAC3 fixed the hyper-parameter values for all individuals and iterations, this is less flexible for particles to exploration and exploitation in the searching space, while DMSPSO itself has effective adaptive rules which ensure a successful searching.
> - As also pointed out by the reviewer, GLEET has the unique advantage that the hyper-parameter values can vary for individuals and for iterations, which is also a reason why GLEET outperforms SMAC3.
>
> We have incorporated these comparison results into Table 6 and the corresponding analysis into Section 5.3.5 of our revised paper.
>
> |  | DMSPSO | SMAC3-DMSPSO | GLEET-DMSPSO |
> | --- | --- | --- | --- |
> | Mean(Std) | 4.771E+02(3.345E+01) | 5.134E+02(3.406E+01) | 4.681E+02(3.650E+01) |
>
> > Otherwise, the other comparisons done in the paper is not quite fair as only the proposed approach variants perform pre-training on a training problem instances.
> >
>
> We agree that adding pre-training HPO schemes such as SMAC3 into comparison makes our benchmark more solid. However, we would clarify that in our Comparison Analysis (Section 5.2), **all RL-based baselines perform pre-training on the training set**, including RL-PSO, RLEPSO, DE-DQN, DEDDQN, LDE and RLHPSDE. We apologize for causing your misunderstanding and would like to recall that the major contribution in this paper is a state-of-the-art RL-based BBO framework, with remarkable generalization ability and interpretability (compared to other RL-based works). The comparison results shown in Table 1 to Table 4 suggest that GLEET can significantly boost the backbone optimizer (at least $28\%$) and achieve state-of-the-art generalization performance both for unseen problems within the same class and for unseen realistic problems (Table 6), across different optimization horizons (Figure 5), population sizes (Figure 4) and problem dimensions (Figure 3).

---

> > ### Author Response · Authors · 2023-09-12
> > **Response#2 to Reviewer #SVNG**
> >
> > > Moreover, for adaptive variants, one could run these variants on training problems, then transfer the results to the test problem, possibly as the initial hyper-parameter values and to limit the range of the hyper-parameter values. In this way, the comparison between adaptive variants and learning-based variants becomes fairer.
> > >
> >
> > We agree with the idea that the hyper-parameter values of adaptive variants on one problem instance can be manually transformed into another instance if proper rules are used and trial and error are conducted, which is quite an interesting research direction for boosting the generalization of existing adaptive variants. However, we believe that the common practice of comparing learning-based methods with traditional adaptive variants is still directly running all of them without considering manually boosting the adaptive variants on the test set, where the same comparison setup is followed by almost all existing RL-based related works and other excellent learning-based BBO works such as RNN-Opt (https://proceedings.mlr.press/v70/chen17e/chen17e.pdf, ICML, 2017), LOIS (https://openreview.net/forum?id=rke8aHHlIH, NeurIPS, 2019), Meta-ES (https://openreview.net/forum?id=mFDU0fP3EQH , ICLR, 2023). We also would like to clarify that the scope of this paper is the generalization and interpretability of RL-based BBO framework and we may not focus on methods to transfer hyperparameter values or boost generalization for those adaptive methods. Our GLEET serves as a pioneering work to leverage transformer architecture to control the exploration and exploitation of backbone optimizer, which we believe would help the development of the research directions of learning-based BBO frameworks.
> >
> > > Limitations 2. Experimental Setting. This is the most critical point. The authors create training problem instances and test problem instances from the SAME function by applying different translation and rotation.
> > >
> >
> > We agree that the training set is augmented by adding different shifts and rotations to the same function. However, we would like to clarify that the random shift and rotation will indeed significantly change the optimization properties of the original function, leading to a more challenging one. This is also one of the primary reasons that the well-known BBO benchmark CoCo and IEEE CEC competitions provide problem instances with rotation and shift to examine the robustness of the benchmarked optimizer. Here we demonstrate the significance of the hardness of the rotations and shifts by testing some manually designed adaptive optimizers on the original 10D function $f_2$ (Schwefel function) in CEC and two transformed sets by adding rotation and shift to those functions with two different random seeds. The results (10 independent runs for each function) are shown below:
> >
> > |  | DMSPSO | GLPSO | MadDE | NL-SHADE-LBC |
> > | --- | --- | --- | --- | --- |
> > | original | 3.354E+00(1.375E+00) | 2.986E+00(1.711E+00) | 1.876E-08(9.244E-09) | 1.334E-08(1.025E-08) |
> > | transform-1 | 3.567E+02(1.667E+02) | 3.031E+02(1.572E+02) | 3.173E+02(1.434E+02) | 3.095E+02(1.531E+02) |
> > | transform-2 | 4.341E+02(1.718E+02) | 5.676E+02(1.698E+02) | 3.577E+02(1.493E+02) | 3.492E+02(1.539E+02) |
> >
> > The results above suggest that: 1) adding rotation and shift to a function will extremely upgrade the hardness of optimization (original vs. transform). 2) adding different rotation and shift to the same function is not isometric, the resulting functions are different for a black-box optimizer.
> >
> > We also notice that in the latest proposed work Meta-ES (https://openreview.net/forum?id=mFDU0fP3EQH , ICLR, 2023), the authors also construct a problem set from CoCo benchmark with random shifts. They meta-learn on this set and then directly meta-generalize to realistic tasks. We would also clarify that although some adaptive variants can handle the changed optimization properties from random rotation and shift, their adaptive rules still require quite expertise and knowledge even exhausted grid fine-tuning. This is the motivation of GLEET: trained on easy-to-construct synthetic distribution, then generalized to unseen realistic tasks. Notably, The results in Table 1 to Table 4 demonstrate that GLEET can enhance the backbone adaptive variants by at least $28\%$ on the test set including various rotation and shift on the same function.

---

> > > ### Author Response · Authors · 2023-09-12
> > > **Response#3 to Reviewer #SVNG**
> > >
> > > > The test and training instances are too similar and such a similar training problem instance is hardly prepared in practice as the target problem is black-box.
> > > >
> > >
> > > We apologize for the misunderstanding for that.  We agree with you that in some realistic scenarios, we do not know the mathematical form of the target task. However, we would like to clarify that once the GLEET’s target is pre-trained on the easy-to-construct synthetic problem sets,  it can be directly applied to unseen realistic tasks without further training. To demonstrate this, we have added a zero-shot generalization study generalization study on Protein-docking benchmark in which we directly test the trained GLEET (trained on 10D $f_{mix}$ set of our CEC train sets, containing all $10$ problem types, 128 samples) on unseen Protein-docking benchmark ($280$ different tasks) in Section 5.3.5 of our revised paper. The corresponding results are shown in Table 6, which suggests that training GLEET on a group of synthetic problems is sufficient for generalizing to unseen realistic scenarios. Nevertheless, for other real-world application scenarios where some historical optimization problem instances do exist, the users may directly augment such instances with shifts and rotations to form a large training set, where the users can enjoy even better in-distribution performance of the RL-based GLEET framework.
> > >
> > > > It is not at all surprising that the good hyper-parameter values are the same for the test and training problem instances as they are isometric. The authors must use the different set of problems as the test and the training problem instances. Use CEC testbed as training, and some real-world applications as testing, for example.
> > > >
> > >
> > > As stated above, we have followed your suggestion and added a zero-shot generalization study on a challenging real-world application: Protein-docking benchmark. We hope this additional comparison and our response above will clear your concerns.
> > >
> > > > Limitation 3. Baseline.
> > > >
> > > >
> > > > As baseline approaches, the authors employ PSO and DE, both of which requires parameter tuning depending on the problem instances. Therefore, the hyper-parameter tuning scheme is required. However, a very natural question arises as how much the proposed approach gains over quasi-hyperparameter-free approaches such as IPOP-CMA-ES and PSA-CMA-ES at the cost of possibly non-negligible training time? The time (and the number of function evaluations) for the training must be reported and discuss whether the performance gain over the baselines and other SOTA approaches such as CMA-ES are significant for spending the training time. Please include quasi-hyperparameter-free approaches such as IPOP-CMA-ES and PSA-CMA-ES as the comparison baseline to show the usefulness of the proposed approach.
> > > >
> > >
> > > We agree with the reviewer that adding quasi-hyperparameter-free approaches (e.g., ES family) as baselines could further highlight the contributions and usefulness of our GLEET. Following the suggestion, we have included them as baselines in the added comparison study of the protein-docking benchmark. We would like to note that in the protein-docking benchmark, all problem tasks pose challenging landscape properties to black-box optimizers. Besides, the seraching space of these tasks is $[-inf,inf]^{12}$ which is extremely larger than that in our CEC benchmark ($[-100,100]$) and well-known CoCo benchmark ($[-5,5]$). The latter is always used to test ES algorithms. We test all baselines in our paper and add CMA-ES, IPOP-CMA-ES and PSA-CMA-ES as new baselines to this generalization study. For all learning-based baselines including our GLEET, they are trained on the 10D $f_{mix}$ set (including all $10$ problem types in CEC, $128$ samples).  The function evaluation for training is set as $10^6$. For CMA-ES, IPOP-CMA-ES and PSA-CMA-ES, we reproduce them by referring to the source codes in libcmaes (https://github.com/CMA-ES/libcmaes), with the corresponding configuration as default. We test each baseline for 10 independent runs and report the average optimization performance with standard deviation in Table 6 of our revised paper. Besides, following your suggestions, we also report the average run time for each baseline on the testbed in the table. We showcase the results of GLEET-DMSPSO and the three ES baselines as follows:
> > >
> > > |  | CMA-ES | IPOP-CMA-ES | PSA-CMA-ES | GLEET-DMSPSO |
> > > | --- | --- | --- | --- | --- |
> > > | Mean(Std) | 4.743E+02(5.972E+01) | 4.716E+02(4.634E+01) | 4.720E+02(4.943E+01) | 4.681E+02(3.650E+01) |
> > > | Runtime (s) | 4.834 | 23.451 | 15.672 | 2.515 |

---

> > > > ### Author Response · Authors · 2023-09-12
> > > > **Response#4 to Reviewer #SVNG**
> > > >
> > > > The results indicate that: 1) Once trained on the CEC testbed, GLEET can be directly applied to unseen realistic tasks without further fine-tuning. 2) The generalization performance of GLEET is superior to ES baselines, with a shorter inference time. 3) IPOP-CMAES performs best against the other two ES baselines but is 4 times slower than CMA-ES.  We agree that the training time for GLEET and other RL-based methods is non-negligible (10h for GLEET). However, the motivation behind this is to learn an adaptive optimizer that could generalize to unseen tasks without labour-intensive fine-tuning (ES algorithms also require hours or even days time to conduct manual tuning to get better performance). We hope our work can help the development of the learning-based line of research.
> > > >
> > > > > Limitation 4. Clarity.
> > > > >
> > > > >
> > > > > Differences from the existing approaches are not clearly stated in the related works section. The authors commented in the introduction “Existing works stipulated DRL to be conducted online, where they trained and tested the model directly on the target problem instance, that is, their methods require (re-)training for every single problem.” In this sentence “existing works” includes all the paper cited in Section 2? Does it include DRL-PSO and RLEPSO? The statements should be clarified.
> > > > >
> > > >
> > > > Sorry for the confusion. We agree that “Existing works stipulated DRL to be conducted online...” is not fully accurate. For existing RL-based methods, some of them need to be conducted online, including  DRL-PSO, DE-DDQN, DE-DQN, RLHPSDE. For the RLEPSO, at each epoch, it randomly chooses a function from its benchmark (CEC 2013 competition benchmarking problems, $28$ in total) for training. The training finishes after n epochs. Then the trained DDPG agent is tested on the same $28$ benchmark functions. However, such a train-test process may be unreasonable and can be regarded as online to some extent. For LDE, the training set is functions from CEC 2013 competition benchmarking problems, and the testing set is CEC 2017 competition benchmarking problems. This offline setting is reasonable to learn in a generalizable way. We have revised all lines involving the corresponding description in our revised paper and carefully checked other statements.
> > > >
> > > > > Limitation 5. Generalization.
> > > > >
> > > > >
> > > > > The authors investigated the generalization ability of the proposed approach in Section 5.3. However, the benchmark problems used in this investigations are all multimodal problems. Therefore, they are all similar. For such problems, I suspect that adequate hyper-parameter values are all similar. Therefore, not only the proposed approach but also existing approaches could perform well. Because the authors claim in the introduction that the generalization ability is the issue of the existing approaches, comparison with existing approaches must be included in this experiments. Moreover, to evaluate generalization ability, one should construct a bit more different sets of test and training scheme.
> > > > >
> > > >
> > > > Regarding the concern about all benchmark problems used being similar, we would like to clarify that: 1) as we replied in Limitation 2, adding rotation and shift can make a significant difference in the optimization properties of that problem (in this case, the $f_2$ Schwefel function) and upgrade the hardness for solving the transformed one. 2) The quite large standard deviations of optimization performance on $f_2$ shown in Table 1 to Table 4 demonstrate that the baselines behave differently in different instances, which serves as circumstantial evidence that these instances are largly different. Hence, the settings can reflect the generalization performance in our view. Nevertheless, we agree with you that we should construct a bit more different sets for comprehensively testing the generalization ability of GLEET. Following your suggestion, we have added a zero-shot generalization study on a challenging realistic benchmark: Protein-docking benchmark. The instances in this benchmark are either unimodal or multimodal and with rugged searching surfaces. The added study is located in Section 5.3.5 in our revised paper. The comparison results with other RL-based baselines and traditional/adaptive PBO baselines are illustrated in Table 6. The results show that GLEET trained on CEC testbed achieved remarkable generalization performance on the totally different Protein docking tasks.

---

> > > > > ### Author Response · Authors · 2023-09-12
> > > > > **Response#5 to Reviewer #SVNG**
> > > > >
> > > > > As for the suspicion that “adequate hyper-parameter values are all similar”, it may not be true.
> > > > >
> > > > > To demonstrate that, we leverage SMAC to optimize the hyper-parameter values of DMSPSO on a training set of $f_1$ (unimodal BentCigar function) and $f_2$ (multimodal Schwefel function). Then we fix the optimized hyper-parameter values for DMSPSO and test it on the corresponding test set (1024 same type function with different rotation and shift). The optimization results are listed below, which indicates that similar parameter values (even identical in this case) can not gain better performance although the problems in the train set and test set are with same type and modal property. We hope these results would clear your doubts.
> > > > >
> > > > > |  | Func1 (unimodal) | Func2 (multimodal) |
> > > > > | --- | --- | --- |
> > > > > | Train set | 9.576E+04(1.171E+05) | 3.712E+02(2.376E+02) |
> > > > > | Test set | 3.677E+05(6.216E+05) | 4.836E+02(2.578E+02) |
> > > > >
> > > > > > The authors claim that the oversimplified state representation and network architecture are the cause of the inefficacy of the existing learning-based approaches in the introduction. However, no empirical or theoretical evidence has been provided in the paper. Please provide some evidence, possibly by adding some ablation study.
> > > > > >
> > > > >
> > > > > We agree with you that empirical/theoretical evidence should be provided to demonstrate the importance of RL-based methods. However, we notice that we have conducted such an ablation study on the network architecture of our GLEET to showcase the importance of network design for the efficacy of RL-based methods in Table 8 of the paper. The conclusion there is that the MHA encoder and the EET decoder in GLEET contribute to the final performance, and the fully-informed MHA encoder plays a crucial role in facilitating the learning of more useful features through self-attention among individual embeddings of the population.
> > > > >
> > > > > For the ablation on the state representation, following your suggestion, we have added an ablation study on the state representation of GLEET in Section 5.5.3 of our revised paper to probe which part of the state feature contributes most to GLEET. Noticing that there are three parts of feature in the total state representation: $s_1 \sim s_4$ denote the search progress of the optimization process, $s_5 \sim s_6$ profile the distribution of population’s objective values, $s_7 \sim s_9$ profile the distribution of candidate solutions in the population.  We first prepare four GLEET-PSO agents: 1) original version. 2) ablate $s_1 \sim s_4$ from the state representation, denoted as w/o $s_1 \sim s_4$. 3) ablate $s_5 \sim s_6$ from the state representation, denoted as w/o $s_5 \sim s_6$. 4) ablate $s_7 \sim s_9$ from the state representation, denoted as w/o $s_7 \sim s_9$. We train the four agents on 10D function train sets and then test them on the corresponding test set for $10$ independent runs. We report the optimization performance of each version of the agent in Table 9 of our revised paper and as below. The results show that: 1) removing any one of the three parts features would significantly affect the learning effectiveness of GLEET. 2) the optimization progress feature $s_1 \sim s_4$ contributes most to GLEET, which can be interpreted as an informative signal telling GLEET when and where to adjust the hyper-parameter values for better searching behavior. 3) a comprehensive state representation would help learning indeed.
> > > > > |  | Func1 | Func2 | Func3 | Func4 | Func5 |
> > > > > | --- | --- | --- | --- | --- | --- |
> > > > > | w/o $s_{1\sim4}$ | 3.546E+06(6.675E+06) | 6.672E+02(2.965E+02) | 2.412E+01(6.153E+00) | 1.745E+00(1.331E+00) | 2.163E+03(2.974E+03) |
> > > > > | w/o $s_{5\sim6}$ | 3.376E+06(5.975E+06) | 6.034E+02(2.311E+02) | 2.358E+01(6.274E+00) | 1.773E+00(1.231E+00) | 2.189E+03(2.112E+03) |
> > > > > | w/o $s_{7\sim9}$ | 3.383E+06(5.668E+06) | 5.977E+02(2.232E+02) | 2.316E+01(5.985E+00) | 1.786E+00(1.187E+00) | 2.107E+03(2.313E+03) |
> > > > > | GLEET | 2.748E+06(4.205E+06) | 5.105E+02(1.776E+02) | 2.120E+01(4.705E+00) | 1.422E+00(7.776E-01) | 1.847E+03(2.552E+03) |
> > > > > |  | **Func6** | **Func7** | **Func8** | **Func9** | **Func10** |
> > > > > | w/o $s_{1\sim4}$ | 7.329E+01(4.993E+01) | 7.588E+02(7.464E+02) | 5.971E+02(3.699E+02) | 3.610E+02(6.631E+01) | 2.876E+02(4.273E+01) |
> > > > > | w/o $s_{5\sim6}$ | 6.912E+01(6.134E+01) | 7.331E+02(7.762E+02) | 2.062E+02(2.371E+02) | 3.034E+02(6.237E+02) | 2.766E+02(4.130E+01) |
> > > > > | w/o $s_{7\sim9}$ | 6.812E+01(6.900E+01) | 7.201E+02(6.861E+02) | 4.691E+02(2.743E+02) | 2.981E+02(6.217E+01) | 2.537E+02(3.935E+01) |
> > > > > | GLEET | 4.449E+01(3.381E+01) | 4.977E+02(3.549E+02) | 1.096E+02(3.924E+01) | 1.665E+02(6.137E+01) | 1.882E+02(4.127E+01) |

---

### Review · Reviewer_zC3L · 2023-08-28

**Summary Of Contributions:**

This paper proposes a new learning based approach (GLEET) to adjust the Exploration-Exploitation Tradeoff (EET) in Population-Based Optimization (PBO). Specifically, it trains a Transformer-based agent with PPO to output the hyper-parameters of a few backbone PBO algorithms dynamically, including PSO, DMSPSO, and DE. It designs the state presentation to be generic to PBO problems so that the same model can be applied across various backbone algorithms. It's trained on a set of synthetic functions and tested on functions in (1) the same family with different input shift and rotation; and (2) different function families. It shows empirically better performance than PBO methods with static, dynamic, and learned EET controls. It also conducts generalization evaluation on different problem dimensions and population size from the training setting and ablation with different reward and network architectures.

The main contribution of this paper compared to existing RL-based EET is the transformer based network. The proposed input state feature presentation and the reward function are also new in the PBO literature AFAIU.

**Audience:**

Yes

**Claims And Evidence:**

No

**Requested Changes:**

Addressing the weaknesses and comments listed above, especially W1 and W3 are critical to evaluate the quality of the proposed algorithm.

**Strengths And Weaknesses:**

Strengths:
1. Detailed literature review and clear presentation of the method.
2. A new transformer-based architecture to output the EET hyperparameters.
3. Superior performance of the proposed GLEET method compared to a variaty of baselines
4. Sufficient experimental analysis of the learned behavior and ablation

Weaknesses:
1. I have some concerns on the results of baselines that lead me to wonder if the baselines are properly setup
  - sDMSPSO (Liang et al., 2015) is comparable or even worse than DMSPSO, in contrast to the description in the related work section "efficiently adjust EET, achieving superior performance"
  - All DRL-based baselines are significantly worse than adaptive EET methods in all table 1-4.

  This lead me to suspect that either the baseline algorithms are not set up / trained properly, or the problem setting in this paper is significantly different from the settings in the original references.

2. More discussion would be useful to explain the advantage of the proposed method over other DRL-based methods. While there are some brief explanation in the related work section, I think the difference deserves more detailed experimental analysis since the proposed Transformer-based architecture (with PPO) is the main methodology contribution of this work.

3. Generalization evaluation is somewhat limited.
  - The most detailed comparison is done for test functions of the same family. While the results are useful,  this is not the real test setting in practice. A user would usually not have access to >100, or any at all, similar functions of the same family for training. The zero-shot generalization is much more valuable in my opinion. Unfortunately, the generalization analysis is done in a limited setting: a single backbone, PSO, on a single training function familiy f2. This is not sufficient to show convincing evidence of generalization
  - Minor comments: in 5.3.2 the fact that applying GLEET trained with 100 population to test with 500 population doesn't make any gain suggest GLEET does not learn how to make use of a bigger population size, which is a somewhat negaive result.
  - Another interesting generalization analysis to do is to run GLEET for more generations than the training horizon. This is a well-know challenge for meta-learned optimizers.

Other comments:
1. Missing references on transformer-based population-based optimizer
  - Lange, Robert, et al. "Discovering Evolution Strategies via Meta-Black-Box Optimization." The Eleventh International Conference on Learning Representations. 2022.
  - Lange, Robert, et al. "Discovering Attention-Based Genetic Algorithms via Meta-Black-Box Optimization." Proceedings of the Genetic and Evolutionary Computation Conference. 2023.

2. The way of using EET as query and population feature as KV in the decoder is somewhat uncommon. A more standard design would be letting the N population embedding as query to attend to EET features as KV. Alternatively, a simpler network design would be passing all the 2N+1 embeddings in the blue box to one encoder and use to last N outputs to compute the prediction heads, that is, omitting the upper right yellow box. I wonder if the authors could comment on this.

3. I would appreciate more discussion on the limitations of the current work. The brief discussion in the conclusion can be further expanded for readers to understand the application scope of the proposed method.

---

> ### Author Response · Authors · 2023-09-12
> **Response#1 to Reviewer #zC3L**
>
> We sincerely appreciate your valuable feedback, and we thank you for acknowledging that our experiments are sufficient and that our proposed GLEET has achieved superior performance compared to a variety of baselines. We have carefully addressed each of your comments and provided a point-by-point response below.
>
> > I have some concerns on the results of baselines that lead me to wonder if the baselines are properly setup
> >
> > - sDMSPSO (Liang et al., 2015) is comparable or even worse than DMSPSO, in contrast to the description in the related work section "efficiently adjust EET, achieving superior performance"
> > - All DRL-based baselines are significantly worse than adaptive EET methods in all table 1-4.
> >
> > This lead me to suspect that either the baseline algorithms are not set up / trained properly, or the problem setting in this paper is significantly different from the settings in the original references.
> >
>
> We understand your concerns regarding the results of the baselines. However, we would like to clarify that most of the results were tested based on the original source code provided by the original authors (including baselines GLPSO, MadDE, NL-SHADE-LBC, LDE, DE-DDQN); for other baselines (including DMSPSO, sDMSPSO, DRL-PSO, RLEPSO, DE-DQN and RLHPSDE) that do not have open-source code available, we implement them strictly following the pseudocodes in their original manuscripts. For all baselines, we have followed their recommended hyper-parameter settings and ensured that the code and settings we used could achieve similar performance on the benchmark they used in their original paper. For example, for our implemented sDMSPSO baseline, we test our implementation under the original heperparameter settings and the benchmark setting (CEC 2015),  we list the performance gap between our implementation and the results reported in the original paper in the table below. The results show that our implementation achieves similar performance, which endorses the fairness of our experiments.
>
> | Func1 | Func2 | Func3 | Func4 | Func5 |
> | --- | --- | --- | --- | --- |
> | 0.1% | -0.9% | -1.1% | 0.4% | 0.1% |
>
> We have carefully added a declaration about the above implementation details in the revised Section 5.1.2.
>
> Regarding the specific concerns about the results, we respond as follows:
>
> > sDMSPSO (Liang et al., 2015) is comparable or even worse than DMSPSO…
> >
>
> Sorry for the confusion. While it is true that sDMSPSO outperforms DMSPSO on the CEC2015 benchmark (as reported in their original paper and reproduced by us), this superiority does not necessarily hold across our more diverse and intricate test instances that contain thousands of instances with different rotation and shift variations (resulting in significantly different landscape properties). A potential reason for this discrepancy is that the manually designed adaptive mechanisms in sDMSPSO might be overly tailored and biased to a limited set of test problems, such as the 15 functions in CEC2015. This suggests that while adaptive versions may excel in specific scenarios after trial and error, they might not always generalize well to other new problems. On the other hand, the original non-adaptive versions might sometimes even exhibit greater robustness. Note that this observation also aligns with our motivation to move away from hand-crafted adaptive rules and to propose the GLEET which explores a learning-based policy network to address the EET issue. Our GLEET minimizes the need for labor-intensive algorithmic designs by deriving them directly from data, which has been shown to yield the best performance.
> > All DRL-based baselines are significantly worse than…
> >
>
> Firstly, we would like to note that the performance of DRL baselines closely matches the results reported in their original publications. They indeed demonstrate the capability to surpass their backbone algorithms, though they still face challenges when compared to state-of-the-art adaptive PBO variants. For instance, in their original paper, LDE was bested by the classic DE algorithm jSO on the CEC2013 benchmark. Such limitations of DRL baselines might be because these methods are designed to train and test on a small set of problems and lack the ability to capture the knowledge in a large training set and generalize it to a large testing set. We note that DRL-based black-box optimization is a field still in its formative stages (it has only been developed for around 5 years). while adaptive PBO methods have enjoyed extensive research over several decades, leading to the evolution of many formidable variants. Our proposed GLEET, nevertheless, significantly narrows this performance gap and has showcased unique advantages and superior performance compared to all the baselines.

---

> ### Author Response · Authors · 2023-09-12
> **Response#2 to Reviewer #zC3L**
>
> > More discussion would be useful to explain the advantage of the proposed method over other DRL-based methods. While there are some brief explanation in the related work section, I think the difference deserves more detailed experimental analysis since the proposed Transformer-based architecture (with PPO) is the main methodology contribution of this work.
> >
>
> We provide additional remarks about the novelty of GLEET below and re-organize part of them into the revised paper, Related Works (Section 2.2) and Conclusion (Section 6). We also revise some statements in each subsection of Experiments (Section 5) to highlight the correspondence between the novelty and the concrete experiment design.
>
> 1. More comprehensive MDP: As mentioned in Section 4.1, we formulate the controlling of EET as a more comprehensive Markov Decision Process (MDP) than those in the existing works. The recent RL-based methods have more or less shortness in their MDP designs. For example: 1) the state representation in RLEPSO is simply a time-stamp $t$ ; the state representation in RL-PSO is the coordinates of solutions and the corresponding objective values $\{X,Y\}$. These state representations may easily get their agents overfited. 2) the action space in DEDDQN and DE-DQN is a discrete space with $4$ or $6$ different mutation strategies. The limited update parameters may affect the Q-net agent searching for optimal policy.  3) the reward functions in some recent RL-based methods require the optimal of the problem to be known (e.g., $R_3$ in DEDDQN), which is actually not accessible for black-box problems. Our GLEET attempts to refine the MDP design by incorporating the EET feature into the Fitness Landscape profiling of the population state representation (comparing the performance of GLEET and DEDQN, which also Fitness Landscape Analysis but without our EET design); providing continuous control for more flexible policy space (comparing the performance of GLEET and DEDDQN in Table 1 and Table 4 ); proposing an easy-to-compute reward function without knowing the optimal (see Ablation study on reward functions in Table 6).
> 2. Transformer-styled architecture: We would clarify that the major motivations for leveraging Transformer-styled architecture are: 1) the order-invariant of the Attention module naturally ensures the output EET control parameters for a particle is invariant to its order in the population. 2) we use self-attention in our Fully Informed Encoder to smoothly encode each individual with population information, serving as a more informative feature map for subsequent decoding. 3) the hyper-parameters of the backbone optimizer are conditioned on the cross-attention (Exploration-Exploitation Decoder) between the EET feature map and the previously self-attended population feature map.  From the results shown in Table 7 of the Ablation study (Section 5.5.2), we observe that both the encoder and decoder designs contribute to GLEET.
> 3. In-depth Interpretability:  compared with other RL-based baselines, GLEET firstly provides a crystal clear way to interpret what has been learned during the large-scale training, which is showcased in Section 5.4: 1) we first illustrate how GLEET dynamically controls the hyper-parameters changes as the optimization process advances in Figure 7. The results show that GLEET would pay more rounds of exploration and exploitation to search for optimal in difficult problems. 2)  GLEET benefits exactly from the Transformer-styled architecture for visualizing when and where the agent determines using exploration or exploitation hyper-parameters for searching optimal (Figure 8). By visualizing the attention weights for particles, we can analyze which particle affects the exploration and exploitation behavior most. The results in Section 5.4 are a novel way to interpret the RL system, which is also an interesting contribution to this paper.

---

> ### Author Response · Authors · 2023-09-12
> **Response#3 to Reviewer #zC3L**
>
> > Generalization evaluation is somewhat limited.
> >
> > - The most detailed comparison is done for test functions of the same family. While the results are useful, this is not the real test setting in practice. A user would usually not have access to >100, or any at all, similar functions of the same family for training. The zero-shot generalization is much more valuable in my opinion. Unfortunately, the generalization analysis is done in a limited setting: a single backbone, PSO, on a single training function familiy f2. This is not sufficient to show convincing evidence of generalization
> > - Minor comments: in 5.3.2 the fact that applying GLEET trained with 100 population to test with 500 population doesn't make any gain suggest GLEET does not learn how to make use of a bigger population size, which is a somewhat negaive result.
> > - Another interesting generalization analysis to do is to run GLEET for more generations than the training horizon. This is a well-know challenge for meta-learned optimizers.
>
> > The most detailed comparison is done for test functions of the same family. While the results are useful, this is not the real test setting in practice. A user would usually not have access to >100…
> >
>
> We agree with the reviewer that in some realistic scenarios, it is hard to augment the task set to be very large, although such training data augmentation may still be feasible if some historical optimization problem exists in other scenarios. Nevertheless, we would like to clarify that the convenience of GLEET is generalizing to realistic tasks with a sufficient set of pre-training problems to easily be augmented. The CEC benchmark (also CoCo used in Meta-ES) provides synthetic functions with different landscape properties, which we believe is the most basic feature shared across black-box optimization problems. To further demonstrate GLEET’s generalization ability, we additionally conduct a zero-shot generalization study on Protein-docking tasks in which the GLEET trained on the $f_{mix}$ set (including all 10 problem types in CEC, 128 samples) is directly used to optimize $280$ challenging realistic Protein-docking tasks. These tasks are expensive to evaluate, hence the $maxFEs$ is set to 1000 and the population size is set to $5$. They also have an extremely complicated search surface and the search space is $[-inf,inf]$.  We provide the results on PSO variants and our GLEET-100pop below, which demonstrate the generalizability of GLEET. We have added these results (including DE variants and our GLEET-DE)  and the corresponding analysis in Table 6 and Section 5.3.5 of our revised paper.
>
> | PSO | DMSPSO | sDMSPSO | GLPSO | DRL-PSO | RLEPSO | GLEET-PSO | GLEET-DMSPSO |
> | --- | --- | --- | --- | --- | --- | --- | --- |
> | 5.153E+02(2.151E+01) | 4.771E+02(3.345E+01) | 5.101E+02(3.155E-01) | 5.061E+02(3.700E+01) | 5.180E+02(4.940E+01) | 4.895E+02(4.129E+01) | 4.832E+02(3.865E+01) | 4.681E+02(3.650E+01) |
>
> > Unfortunately, the generalization analysis is done in a limited setting: a single backbone, PSO, on a single training function familiy f2…
> >
>
> Following your suggestions, we now present the complete dimension and population size generalization results (all problem types in CEC) of GLEET in the new Figure 3 and Figure 4. Results in Figure 3 show that on approximately half of problem types, GLEET trained on 10D problems can achieve similar performance with GLEET trained on 30D problems. Results in Figure 4 show that GLEET trained with a smaller population size (100) can achieve similar or even better performance than the GLEET trained with a larger population size (500). Both results demonstrate that GLEET has a good generalization ability across different population sizes as well as different problem dimensions.
> > Minor comments: in 5.3.2 the fact that applying GLEET trained with 100 population to test with 500 population doesn't make any gain…
> >
>
> Sorry for causing your misunderstanding about this. We would like to clarify that GLEET-500pop is trained with a 500 population size and GLEET-100pop with a 100 population size. Considering the same number of function evaluations, GLEET-100pop will involve more generations than GLEET-500pop, this is the major reason why the result looks like GLEET-500pop doesn’t make any gain. However, we also provide the reference line of PSO-500pop to demonstrate GLEET-500pop indeed makes some gains. We would also clarify that the surprising result (GLEET-100pop > GLEET-500pop) does not always happen, and we have revised our paper and provided all population generalization results (all 10 problem types in CEC) in Figure 4. The results show that GLEET-100pop may surpass GLEET-500pop on some problem types. This can be regarded as an advantage of GLEET since one can directly apply GLEET with a smaller population size to achieve acceptable results, saving both computational resources and time.

---

> ### Author Response · Authors · 2023-09-12
> **Response#4 to Reviewer #zC3L**
>
> > Another interesting generalization analysis to do is to run GLEET for more generations
> >
>
> We also conduct a generalization analysis of running GLEET for more generations. Due to the restrictions on time (2 weeks due), we take GLEET-DMSPSO in the $f_2$ problem as an example. Figure 5 in Section 5.3.3 shows that GLEET retains outstanding performance with more generations.
>
> > Missing references on transformer-based population-based optimizer
> >
> > - Lange, Robert, et al. "Discovering Evolution Strategies via Meta-Black-Box Optimization." The Eleventh International Conference on Learning Representations. 2022.
> > - Lange, Robert, et al. "Discovering Attention-Based Genetic Algorithms via Meta-Black-Box Optimization." Proceedings of the Genetic and Evolutionary Computation Conference. 2023.
>
> Thanks for your valuable suggestion. We have added the Meta-ES and Meta-GA as related works in Section 2 of our revised paper (colored in blue). These two works provide a brand new paradigm to meta-learn an NN-parameterized population-based optimizer by Neural Evolution. The training benchmark is CoCo, which is similar to our CEC $f_{mix}$ test set.  Notably, Meta-ES shows remarkable robustness when conducting zero-shot generalization to high-dimensional continuous controlling tasks.
>
> > The way of using EET as query and population feature as KV in the decoder is somewhat uncommon. A more standard design would be letting the N population embedding as query to attend to EET features as KV. Alternatively, a simpler network design would be passing all the 2N+1 embeddings in the blue box to one encoder and use to last N outputs to compute the prediction heads, that is, omitting the upper right yellow box. I wonder if the authors could comment on this.
> >
>
> > The way of using EET as query and population feature as KV in the decoder is somewhat uncommon…
> >
>
> We acknowledge that the way you suggested may also be reasonable. The major consideration why our design is in the opposite way is because the intuitive idea of related RL-based methods is embedding the individual with some landscape features and using this population feature directly to infer the control parameters (e.g., in LDE, they use the histogram logits of each individual’s evolution path as the decision vector). Our design aligns with these works by letting the EET feature of each individual be the query, in a sense that: given the personal best (pBest) and global best (gBest) information of that individual and all other individual population features, what extends that individual should let the other individual’s features attend to its own feature.
>
> > Alternatively, a simpler network design would be passing all the 2N+1 embeddings in the…
> >
>
> Thanks for the question. We would like to clarify that this simpler design may have several drawbacks: 1) complexity: approximately, the complexity for attention with $2N+1$ inputs is fairly larger than two attention modules (encoder and decoder in our design) with $N$ inputs. 2) dilution: the gBest embedding in the EET feature would be diluted during the self-attention among the $2N+1$ embeddings. Nevertheless, we also train your suggestion on the 10D CEC testset $f_{mix}$ (mixture of all $10$ problem types, also $128$ samples) and compare it with our original reported reulst (GLEET-DMSPSO). The suggested structure achieves $1.565E+02$, which is not as good as our design ($1.364E+02$).
>
> > I would appreciate more discussion on the limitations of the current work. The brief discussion in the conclusion can be further expanded for readers to understand the application scope of the proposed method.
> >
>
> Thanks for the suggestion. We now list several potential limitations of GLEET here and have added them in the Conclusion (Section 6) of the revised paper.
>
> - Although GLEET has shown the state-of-the-art generalization performance among the RL-based methods, the handcrafted population features and EET features, which are based on the Fitness Landscape Analysis may still show vulnerability to the high-dimensional scenario.  This situation leaves us an open question: How to design an automatical representation system that accurately profiles the population’s optimization status and hence boosts the downstream ML for Optimization tasks such as GLEET, Meta-ES, etc.
> - To apply GLEET, the backbone optimizer must have explicit EET control parameters, such as $F$ and $C_r$ in DE, $w$ and $c$ in PSO, as well as $\mu$ and $\sigma$ in CMA-ES.
> - To secure the correctness of the MDP, the population size of the backbone optimizer can not change dynamically. If the population size reduces as the optimization process advances (e.g., non-linearly reduced population size in NL-SHADE-LBC), the action space of the MDP changes, hence the training becomes meaningless.

---

### Decision · Action_Editors · 2023-10-05

**Recommendation:** Reject

**Comment:**

As the primary acceptance criterion is having evidence supporting the claims, this paper is not yet ready to be published at TMLR. However, the reviewers have given the authors lot of feedback. This should help improve the paper. I encourage the authors to take these points into consideration and update the paper accordingly.

One minor note: it is quite difficult to understand the magnitudes of the results presented in tables especially when the numbers are expressed in scientific notation (Tables 1-5). I suggest the authors choose a more visually appealing way to show those results, perhaps using bar graphs.

**Audience:**

The audience is appropriate for TMLR.

**Claims And Evidence:**

As pointed out in the reviews, the paper had some weakness. In particular, the significant problems are:

- Statistical significance: results are too noisy, and it is difficult to claim improvements due to remaining withing the standard deviation of the comparison points.

- Generalizability: the problems are drawn from the same family of functions as the training set, and they are all too similar.

- Baselines: there are multiple concerns that the baselines are not properly implemented.

As a result, the several reviewers concluded that the evidence does not support the claims.

---

> ### Author Response · Authors · 2023-10-13
> **Clarification and Request for Guidance**
>
> Dear Action Editor,
>
> First and foremost, we would like to express our sincere gratitude for your dedicated efforts in managing the review process and for the invaluable feedback provided by the reviewers.
>
> However, we are currently facing some confusion regarding the review process. Our paper underwent a comprehensive review by three reviewers by August 28, and we subsequently addressed their comments and made significant improvements, which were uploaded on September 12. These enhancements included, but were not limited to:
>
> 1.	Providing additional results to demonstrate statistical significance.
>
> 2.	Expanding our generalization studies, including a zero-shot generalization study on the challenging Protein-docking benchmark.
>
> 3.	Addressing the implementation issues raised and introducing more baseline methods (SMAC3, CMA-ES, IPOP-CMA-ES, and PSA-CMA-ES) for comprehensive comparison.
>
> For more details, please refer to our responses to the respective reviews listed below.
>
> However, we are perplexed by the fact that we did not receive any further feedback on our updated paper, only to receive a direct decision of rejection. This seems at odds with your suggestion: "However, the reviewers have given the authors lot of feedback. This should help improve the paper. I encourage the authors to take these points into consideration and update the paper accordingly." We believe there may be some information gap or miscommunication in this matter.
>
> Could you kindly provide us with additional guidance regarding the next steps? Please accept our sincere thanks in advance for your assistance.
>
> Best regards,
>
> TMLR Paper1302 Authors